# Chemical Characterization of 29 Industrial Hempseed (*Cannabis sativa* L.) Varieties

**DOI:** 10.3390/foods13020210

**Published:** 2024-01-09

**Authors:** Sheyla Arango, Jovana Kojić, Lidija Perović, Branislava Đermanović, Nadežda Stojanov, Vladimir Sikora, Zorica Tomičić, Emiliano Raffrenato, Lucia Bailoni

**Affiliations:** 1Department of Comparative Biomedicine and Food Science (BCA), University of Padova, Viale dell’Universitá 16, 35020 Legnaro, PD, Italy; sheylajohannashumyko.arangoquispe@phd.unipd.it (S.A.); emiliano.raffrenato@unipd.it (E.R.); 2Institute of Food Technology in Novi Sad, University of Novi Sad, Bulevard Cara Lazara 1, 21000 Novi Sad, Serbia; jovana.kojic@fins.uns.ac.rs (J.K.); lidija.perovic@fins.uns.ac.rs (L.P.); branislava.djermanovic@fins.uns.ac.rs (B.Đ.); zorica.tomicic@fins.uns.ac.rs (Z.T.); 3Institute of Field and Vegetable Crops, Maksima Gorkog 30, 21101 Novi Sad, Serbia; nadezda.stojanov@nsseme.com (N.S.); vladimir.sikora@ifvcns.ns.ac.rs (V.S.)

**Keywords:** hemp, cannabis, hempseed, cannabinoids

## Abstract

Hemp is considered one of the potential novel crops for human and animal nutrition. This study aimed to determine the complete chemical composition of 29 different varieties of whole hempseeds. Fatty acid composition, amino acid profile, mineral composition, and cannabinoids content were also evaluated. All hempseed varieties were milled to obtain whole hempseed flour. Differences between hempseed varieties were significant (*p* < 0.05) for all measured parameters. Proximate composition results showed that crude protein and fat contents varied from 21.6–28.9% and 21.1–35.7%, respectively. Fatty acid profiles revealed that the three major fatty acids were linoleic acid (52.79–57.13%) followed by α-linolenic acid (12.62–20.24%), and oleic acid (11.08–17.81%). All essential amino acids were detected in all varieties, with arginine (12.66–17.56 mg/100 g protein) present in abundance, whereas lysine was limiting. Substantial differences were found in the mineral content, and potassium (509.96–1182.65 mg/100 g) and iron (5.06–32.37 mg/100 mg) were the main macro- and microminerals found. All cannabinoids were found in small traces and tetrahydrocannabidiol (THC) was only detected in five varieties. To conclude, the nutritional composition of hempseeds with hull makes them suitable to be added into the diets of humans or animals as a highly beneficial novel ingredient.

## 1. Introduction

Industrial hemp (*Cannabis sativa* L.) seeds have a beneficial nutritional composition as they are a rich source of protein, unsaturated fatty acids, and some minerals [1]. Even though studies on the chemical composition and nutrient contents of hempseed show significant variation among hemp varieties, which it is known can be directly correlated with the genotypes and environmental factors (rainfall, temperature, soil, etc.), limited data are available. Hempseed contains about 25% protein. A total of 181 proteins have been identified in hempseeds with the main ones being the globular-type albumin (25–37%) and the legumin-type globulin edestin (67–75%) [1]. Hempseed is an excellent source of digestible amino acids as it contains high levels of arginine, aspartic acid and glutamic acid [2]. Concerning fatty acids, the polyunsaturated ones are predominant in hempseeds. Hempseed contains significant amounts of linoleic acid, which accounts for more than half of its total fatty acids. The remaining fatty acid content is composed of α-linolenic acid (16–19%), oleic acid (12–17%), palmitic acid (5–8%), γ-linolenic acid (1–3%), and some other minor fatty acids [3,4]. The total mineral content of hempseed is often reflected by the ash content [1] and the main minerals are calcium, magnesium, potassium, iron, manganese, copper, and zinc. Some differences in mineral content have been reported due to environmental conditions, agricultural practices, and varieties.

Cannabinoids are among the 400 different chemical substances that have been isolated from hemp. They are mostly in the inflorescence of the plant and the most abundant are cannabidiol (CBD) and tetrahydrocannabidiol (THC). CBD is the main non-psycotrophyc cannabinoid, while THC is the only psychoactive component in hemp. Even though industrial hemp has less than 0.2% of THC [5], this is a critical consideration for both consumers and industries because of the potential accumulation of THC in animal tissues and its effect on animal health, production, and food product quality [6,7]. In fact, it has been proved that cannabinoids of hemp by-products can transfer into milk when they are included in dairy cows’ diet [8,9]. Therefore, minimizing the risk of psychoactive effects associated with THC to make industrial hemp seeds safe for consumption needs to be guaranteed by always measuring its content. In this way, consumers and industries can relay this information and confidently incorporate these seeds into various products, such as food items and other nutritional supplements, knowing that they offer valuable nutrients without the concerns associated with high THC levels.

Due to its impressive nutritional profile and bioactive compounds already mentioned, hemp seeds are often considered nutraceuticals because they provide health benefits beyond basic nutrition. The remarkable potential of hempseed as an innovative candidate for both food and feed applications underscores the impetus behind the cultivation of diverse industrial hemp varieties in various countries. The aim of this study was to determine the complete chemical composition and nutritional characteristics of 29 different hempseed varieties, and also to contrast them with those of soybean.

## 2. Materials and Methods

### 2.1. Materials

Whole hempseeds of 29 varieties were used in this study (Table 1), some of them monoecious and some of them dioecious. They are originally from 8 different countries and 7 of them are not registered in the European Union so are not included in the Plant Variety Database of the European Commission [10]. The place of cultivation was the Institute of Field and Vegetables Crops, Department for Alternative Crops, Bački Petrovac, Serbia (45.336500° N 19.671355° E). The soil was alluvial chernozem with a pH of 7.2 and the previous crop was sorghum. The soil preparation consisted of deep plowing, followed by disking and cultivation for the establishment of a suitable seedbed. Before plowing in fall, fertilization with nitrogen, phosphorus and potassium was performed in a proportion of 16:16:16 at 300 kg/ha. The sowing took place on April 2021. Each variety had 3 rows that were 10 m long and the distance between plants was 50 cm. Plots were kept weed-free by mechanical cultivation until the fifth week after emergence. The harvest was performed manually in October 2021.

Soybean rubin cv. was cultivated at the Institute of Field and Vegetables Crops, Department of Legumes, Rimski šančevi, Serbia (45.329146° N 19.835969° E) in 2022. The soil was chernozem with a homogeneous texture and a well-aggregated structure.

Some climatic parameters of the hemp cultivation site like temperature and precipitation (Figure 1) were measured at the meteorological station located approximately 500 m away from the experiment.

### 2.2. Sample Preparation

Industrial hempseeds and soybean with hulls were milled (FOSS KN295 Knifetec, Labtec Line, Hillerod, Denmark) to obtain flour at the Institute of Food Technology (FINS) of the University of Novi Sad, Serbia. All analyses were performed using 2 replicates.

### 2.3. Chemical Composition

Moisture, ash, crude protein (CP) (ISO20483) [11] and lipids (EE) were determined according to the AOAC international standards [12] at the Institute of Food Technology (FINS) of the University of Novi Sad, Serbia. Total carbohydrates (CHO) were calculated by difference in percentages: 100% − (moisture + ash + crude protein + lipids)%.

### 2.4. Fatty Acid Profiles

Preparation of methyl esters of fatty acids and determination by capillary gas chromatography (Agilent 7890A system, Agilent Technologies, Santa Clara, CA, USA) equipped with a flame ionization detector (FID) and a SP-2560 fused-silica capillary column (100 m × 0.25 mm × 0.20 µm film thickness) were performed following official procedures (ISO12966-2, ISO12966-4) [13,14] at the Institute of Food Technology (FINS) of the University of Novi Sad, Serbia. A methyl ester standard mix of 37 fatty acids, Supelco 37 FAME mix (Supelco, Bellefonte, PA, USA) was used as an internal standard for the analysis of each sample. Results were expressed as percentages of total FA.

### 2.5. Amino Acid Profiles

The amino acid composition was determined by ion-exchange chromatography using an automatic amino acid analyzer Biochrom 30+ (Biochrom, Cambridge, UK) at the Institute of Food Technology (FINS) of the University of Novi Sad, Serbia following the method described by Spackman et al., 1958 [15]. The technique was based on amino acid separation using strong cation-exchange chromatography, followed by the ninhydrin color reaction and photometric detection at 570 nm, except for proline, which was detected at 440 nm. L-norleucine (Sigma-Aldrich, St. Louis, MO, USA) was used as an internal standard in each sample analysis.

Samples were initially hydrolyzed in 6 M HCl (Merck, Germany) at 110 °C for 24 h. Similarly, alkaline hydrolysis with 4 M NaOH was used for the determination of tryptophan. After hydrolysis, samples were cooled at room temperature and dissolved in 25 mL of loading buffer (pH 2.2) (Biochrom, Cambridge, UK). Samples were subsequently filtered through a 0.22 μm pore size PTFE filter (Merck KGGaA, Germany) and transferred into a vial (Agilent Technologies, USA) and stored in a refrigerator prior to analysis.

The amino acid peaks were identified by comparison of retention times with the standard purchased from Sigma-Aldrich (Amino Acid Standard Solution, Sigma-Aldrich, St. Louis, MO, USA). The results were expressed as total protein basis (g/100 g protein) or whole seed basis (g/100 g seeds).

### 2.6. Mineral Composition

The mineral composition (Ca, Mg, K, Fe, Mn, Cu, Zn, and Na) was determined by flame-atomic-absorption spectroscopy (AAS) at the Institute of Food Technology (FINS) of the University of Novi Sad, Serbia, according to the official method ISO6869 [16]. This methodology did not allow for the determination of phosphorus content.

### 2.7. Cannabinoid Analysis

The cannabinoid analysis was performed at the Institute of Field and Vegetable Crops of the University of Novi Sad, Serbia, according to the procedure described by Zeremski et al., 2018 [17]. Absolute ethanol (10 mL) was added to 2 g of dried homogenized sample in an Erlenmeyer flask with a stopper. The solution was sonicated for 15 min and then centrifuged at 10,000 rpm for 5 min. The supernatant was transferred into a GC vial. The decarboxylation step of the acidic forms of CBD and THC was achieved in the GC-MS inlet at a temperature of 280 °C. 

Analysis of cannabinoids was performed on an Agilent 6890 N gas chromatograph equipped with a mass-spectrum detector. The separation was performed on a fused-silica capillary column (HP-5, 30 m × 0.25 mm i.d., and 0.25 µm film thickness). Helium was used as a carrier gas at a constant flow of 1 mL/min. The temperature program was as follows: initial temperature of 200 °C was held for 2 min, then increased to 240 °C at a rate of 10 °C/min, and kept for 10 min. The injector and detector temperatures were set at 280 and 230 °C, respectively. The injected sample volume was 1.5 µL and the split ratio was 1:20. Individual analytical standards for cannabidiol (CBD) and cannabinol (CBN) were used for calibration. Quantification of THC was performed with CBN analytical standard in accordance with the method given by Poortman-van der Meer and Huizer (1999) [18] and was expressed as % in dry weight.

### 2.8. Statistical Analysis

All analyses were performed in duplicates. The results were subjected to analysis of variance (ANOVA) by using a completely randomized design method (SAS Inst. Inc., Cary, NC, USA). Pair-wise comparisons were performed using the Tukey test. Significant levels were considered as *p* < 0.05.

## 3. Results

### 3.1. Chemical Composition

Proximate analyses of 29 hempseed varieties are shown in Table 2. Some differences in dry matter (DM), CP, EE, CHO, and ash contents were found (*p* < 0.05). Dry matter among all hempseeds ranged from 90.34 to 93.52%. The lowest value was recorded from the Marina variety and the highest from the Ferimon FR 8194 variety. The crude protein content of hempseeds ranged from 21.63 to 28.92% DM. The lowest value was recorded from the Monoica variety and the highest from the Tisza variety. The fat content of hempseeds ranged from 21.12 to 35.67% DM. The lowest value was recorded from the Epsilon 88 variety and the highest from the Fibrol variety. Total carbohydrate content from hempseeds ranged from 25.49 to 43.00% DM. The lowest value was recorded from the Tisza variety and the highest from the Epsilon 88 variety. Ash content from hempseeds ranged from 4.40 to 7.49% DM. The lowest value was recorded from the Wojko variety and the highest from the KC Virtus variety. The Tisza variety was the most balanced for crude protein and fat content. In comparison to the sample of soybean, the crude protein content is much higher in soybean (41.79 vs. 25.17%) whereas the fat content (18.65 vs. 31.72%) is much lower in soybean than the average of all hempseeds. Similar values of total carbohydrate content were found between soybean and the mean of all hempseed varieties.

### 3.2. Fatty Acid Composition

The FA content (% of FAME) of all the 29 hempseed varieties are presented in Table 3 and Table 4, as well as the chromatograms of fatty acid methyl esters from total fatty acids (Appendix A). The different varieties of hempseeds showed statistically significant differences for all individual FAs and fatty acid groups (*p* < 0.05). The main FAs of hempseeds were linoleic acid (18:2n−6), from 52.79 to 57.13; α-linolenic acid (C18:3n−3), from 12.62 to 20.24; and oleic acid (C18:1), from 11.08 to 17.81.

The hempseeds were characterized by a high proportion of unsaturated fatty acids, from which PUFA represented a high proportion ranging from 70.73 to 71.27. In terms of SFA, it ranged from 10.02 to 13.01. Palmitic acid (C16:0) was the leading FA of this group with a concentration that ranged from 3.79 to 8.13. The Kina and Santhica 23 varieties were the highest in PUFA and lowest in SFA. The ω6/ω3 ratio ranged from 2.86 to 4.71. The lowest value corresponded to the Dioica 88 and Kina varieties, whereas the highest belonged to the Wojko variety.

Palmitoleic (C16:1), γ-linolenic (C18:3n−6), and eicosadienoic acid (C20:2n−6), all of which belong to the group of unsaturated FAs, were all present in hempseeds but not found in soybean. Interestingly, the soybean ω6/ω3 ratio was 9.72, which is more than twice as high as the average of all hempseeds.

### 3.3. Amino Acid Profiles

The amino acid profiles of 29 hempseed varieties are reported in Table 5 and Table 6, expressed as total protein basis (g/100 g protein) or whole seed basis (g/100 g seeds), respectively, as well as all their chromatograms (Appendix A). Some differences in all amino acids were found between hempseed varieties (*p* < 0.05). The essential amino acids identified in hempseeds were: histidine, isoleucine, leucine, lysine, methionine, phenylalanine, threonine, valine, and tryptophan. Among these AAs, hempseed varieties were characterized by a high content of leucine that ranged from 6.79 to 7.97 mg/100 g protein. However, phenylalanine was not detected in some hempseed varieties and its content ranged between 0.01 to 4.26 mg/100 g protein. The non-essential amino acids found were: alanine, arginine, asparagine, cysteine, glutamic acid, glycine, proline, serine, and tyrosine. Among these AAs, hempseed varieties were characterized by high contents of arginine and glutamine that ranged from 12.66 to 17.50 and 13.04 to 15.49 mg/100 g protein, respectively. The content of the sulfur-containing amino acids cystine and methionine ranged from 1.25 to 1.89 and 0.90 to 2.27 mg/100 g protein, respectively. Of all of the hempseed varieties, Fedora 17 and Tisza showed the highest contents of all amino acids.

### 3.4. Mineral Composition

Hempseed mineral composition for some minerals known to be essential in human and animal nutrition are shown in Table 7. Some content differences for all of the mineral contents were found between hempseed varieties (*p* < 0.05). Sodium was analyzed but was not detectable (<0.5 mg/100 g sample) in any of the samples, so it is not presented in the results. Helena was the most outstanding variety in terms of mineral content. Compared to soybean, hempseed varieties showed similar or greater amounts of most of the minerals evaluated except for calcium. Calcium content was much higher in soybean (294.17) than in hempseeds (130.57). Manganese and sodium were the two minerals found to be undetectable in soybean.

### 3.5. Cannabinoid Content

The cannabinoid content (µg/g) of 29 varieties of hempseeds is shown in Table 8, as well as their chromatograms (Appendix A). Some differences in the three types of cannabinoids were found between hempseed varieties (*p* < 0.05). They showed average CBD, THC, and CBN contents (µg/g) of 39.54, 7.83, and 18.31, respectively. CBD was present in all hempseed varieties, while CBN and THC was not detectable in some of them. For the 29 varieties analyzed, all of them were under the 0.02% limit value of THC [5]. In fact, only five varieties contained THC, ranging from 15.21–163.33 µg/g equivalent to 0.0015–0.0163%.

## 4. Discussion

### 4.1. Chemical Composition

In general, the hempseed chemical composition is consistent with previous studies in which the variety and agronomic conditions affected the composition of hempseeds [19,20,21]. The analysis of variance (ANOVA) in this study showed significant differences (*p* < 0.05) in the dry matter, crude protein, ether extract, and ash contents among the 29 hempseed varieties analyzed. Results showed that whole hempseeds contain on average 92.83% of dry matter, which is consistent with previous studies that reported values between 91.2 to 96.21% [22,23,24,25,26,27]. This could be due to environmental and storage conditions differences [20]. The soybean used had a lower dry matter content (89.91%) compared to the 29 varieties of hempseed, probably due to the differences in agronomical practices and storage. Protein content (% of DM) in whole hempseeds was 25.17; this value is high and fits within the wide range of 12.2 to 28 sourced from different studies [19,20,22,23,24,25,26,28,29]. In the case of soybean, which has been always considered a rich source of protein, the crude protein quantity (41.79%) was notably higher. Fat content (% of DM) was the most abundant fraction in whole hempseeds representing an average of 31.72. This value is high if we consider the wide range of 9.31 to 35.00 found in the literature [19,20,22,23,24,25,26,27,28,30] and also in comparison to the fat content found in the soybean (18.65). The ash content (% of DM) of hempseeds was 5.54 and also seems to be consistent with previous works that reported values between 4.4 to 7.20 [19,23,24,26,29,30] and it is close to the content found in soybean (5.20).

The chemical composition of some of the hempseed varieties evaluated were found in the literature as whole seeds, which is why the content of crude protein (Figure 2) and fat (Figure 3) could be compared. Crude protein was higher on comparison to reports from all the other authors. Alonso-Esteban et al., 2022 [27] found lower values for CP in all of the varieties examined mostly because a lower nitrogen–protein conversion factor (5.3) was used. Fat content was almost similar to other authors’ reports. For the Fedora 17 variety, a wide variation was found within the literature.

### 4.2. Fatty Acid Profile

The FA profiles of hempseeds reported in this work are consistent with those of other authors. All of them, and also this current work, reported linoleic acid (C18:2n−6) as the major FA followed by α-linolenic acid (C18:3n−3) or oleic acid (C18:1) [22,25,26,28,30,31,33,34,35,36,37,38]. In addition, all of the studies concluded that FA composition is highly influenced by the genotype [38,39], which is in agreement with this study where significant variation in the FA profile between hempseed varieties was found (*p* < 0.05). Overall, the FA composition of 29 hempseed varieties revealed a distribution of SFA, MUFA, and PUFA of 10.07–12.95, 11.23–17.93, and 70.74–78.17% of FAME, respectively. The values of SFA were higher than the values reported in the literature, whereas the MUFA and PUFA values were similar to previous studies (SFA: 8.24, 8.60, 9.33, and 9.34; MUFA: 9.37, 11.16, 14.56, 16.05, and 18.70; PUFA: 71.98, 72.58, 74.78, 76.4, and 77.7) [25,26,28,34,35]. In comparison to soybean, the FA distribution within hempseeds is higher in PUFA and therefore lower in SFA and MUFA. This was mostly because soybean lacks γ-linolenic and C20:2n−6, not to mention the fact that its content of α-linolenic is almost three-times less than that in hempseeds. The ω6/ω3 ratio of all of the hempseed varieties were less than the 3:1 to 5:1 established by the European Food and Safety Authority (EFSA) that ensures the maintenance of an optimal state of health in humans. In contrast, this ratio (9.72) was much higher in soybean mostly because its low content (5.35) of C18:3n−3.

Individual comparisons of FA profile (% of FAME) could be performed for some of the hempseed varieties studied in previous works (Figure 4, Figure 5 and Figure 6) in which total contents of SFA, MUFA, and PUFA are in accord with to the values reported here [19,31,33,35].

### 4.3. Amino Acid Profile

First of all, it is evident that the way of expressing the amino acid content needs to be considered in order to make comparisons or state nutritional requirement fulfillments. The way in which the source of protein is going to be assumed by the consumer needs to be specified. Our results showed that hempseed and soybean seem to have a similar amino acid profile when they are expressed in terms of total protein (Table 5), but when it comes to report it in whole seed basis (Table 6) soybean clearly has an advantage because of its higher protein content. So, if the purpose is to use protein isolates as food ingredients it is acceptable to report the amount of amino acid expressed in protein. On the other hand, hempseeds when used as natural food ingredient in a normal based diet are better considered as seeds.

In general, our results are consistent with previous studies in which the variety and agronomic conditions affected the amino acid profile. The amino acid composition of hempseeds showed that they contain all the essential amino acids (EAAs) required for human health as many other authors have already reported [4,20,21,23,30,37,38,40,41]. Hempseed presents very high levels of arginine and glutamic acid and a very low level of lysine [38]. In fact, this study identified arginine (3.64 g/100 g seed) as the main amino acid, followed by glutamic acid (3.26 g/100 g seed). This represents an interesting fact because arginine has been known for its beneficial properties like ammonia detoxification, fetal growth enhancing, and insulin resistance reduction [38].

In comparison to soybean, all EAAs were higher (Figure 7). In fact, lysine would be the first limiting amino acid in hempseeds (2.13 vs. 0.85 g/100 g seed) that has been already stated in the literature [20,42,43,44]. Both hempseed and soybean are similarly low in sulfur-containing amino acids. Considering the average EAA contents of the 29 whole hempseed varieties and a serving size of 30 g, hempseeds fulfilled around 20 percent of a 70 kg adult’s daily requirement suggested by the FAO/WHO (Figure 8) [45].

### 4.4. Mineral Composition

Regarding the mineral content (mg/100 g) of hempseeds, we can agree together with other authors that they are a good source of minerals like potassium, magnesium, calcium, iron, manganese, zinc, and copper [21,38]. Sodium was reported to be less than 5 mg/100 g [46]; nevertheless, it was not detectable (<0.5 mg/100 g) in any the hempseed samples analyzed herein. Other authors also reported high variability in mineral content due to the plant variety [38]. Regarding hempseeds, the two macro elements potassium and magnesium were observed in higher concentrations than other minerals. The average content of potassium was 799.84, which was reported to be the highest [4] or second highest mineral content found in hempseeds [20,28,38] within the range of 250–2821 in the literature [38]. The average contents (mg/100 g) of magnesium and calcium were 370.26 and 130.57, respectively, which is within the ranges of 237–694 and 90–955 reported in the literature for each mineral, respectively [38]. The two micro-elements manganese and copper were found in lesser quantities. The average contents of manganese and copper were 8.41 and 1.09, respectively, which is within the ranges of 4–15 and 0.5–2 reported in the literature for these two minerals [38]. The average content of iron was 12.78, within the range of 4–240 from the literature [38].

Comparing hempseeds to soybean, we can say that hempseeds are a richer source of micro-elements (mg/100 g) such as manganese, copper, and zinc. Manganese was found in hempseeds (8.41) but it was not present in soybean. The amounts of copper and zinc were also higher in hempseeds than in soybean (Cu: 1.09 vs. 0.67; Zn: 4.52 vs. 2.67 resp.). On the contrary, calcium was found in amounts that were twice as high in soybean (294.17) than in hempseeds (130.57).

A few studies regarding the mineral contents (mg/100 g) of whole hempseed for some varieties were found in the literature. For the Fedora 17 variety, the result for magnesium (298.57) in this experiment was consistent with reports in the literature with values of 268.21 and 410.9. Potassium content (854.86) was higher than the values of 251.74 and 709.3 reported. Calcium content (125.47) was within the values of 94.44 and 189.0 found in the literature. Iron content (6.98) was consistent with the given values of 6.45 and 9.80 from other studies. Zinc content (4.47) was also close to the values of 4.84 and 6.69 found in the literature. Manganese content (8.02) was higher than the 4.44 and 6.47 reported in the literature. Copper content (1.21) was within the reported values of 0.50 and 2.76 [3,39]. In the case of the Futura 75 variety, our results showed a calcium content of 99.61, less than the 177.5 found in the literature [40]. Regarding the Carmagnola variety, the content of magnesium (321.14) in this study was a little lower than the reported in the literature of 394.9. Potassium content (1041.25) was higher than the value of 616.7 reported. Calcium content (113.07) was lower than the value of 211.9 from the literature. Iron content (16.58) was higher than the given value of 10.65 from other studies. Zinc content (3.05) was lower than the value of 9.71 found in the literature. Manganese content (8.01) was close to the 9.71 value from the literature. Copper content (1.16) was also close to the reported value of 2.20 [46]. In the case of the Felina 32 variety, the magnesium content (297.43) measured in this experiment was lower than that reported in the literature of 367.1. Potassium content (934.01) was higher than the value of 551.9 reported. Calcium content (133.77) was lower than the value of 181.7 from the literature. Iron content (14.31) was higher than the given value of 7.72 from other studies. Zinc content (3.08) was lower than the value of 7.02 found in the literature. Manganese content (6.12) was slightly lower than the 7.41 from the literature. Copper concentration content (1.00) was lower than the reported value of 2.67 [46]. For the KC Dora variety, the result for magnesium content (499.90) in this experiment was higher than the report of the literature of 365.9. Potassium content (744.45) was also higher than the value of 656.4 reported. Calcium content (119.01) was lower than the value of 161.3 from the literature. Iron content (5.85) was lower than the value of 6.39 given by other studies. Zinc content (5.58) was lower than the value of 7.11 found in the literature. Manganese content (9.41) was higher than the value of 7.34 from the literature. Copper content (1.10) was close to the reported value of 1.63 [39]. For the Kompolti variety, the amount of magnesium (470.11) in this study was higher than the report in the literature of 375.5. Potassium content (652.69) was lower than the reported value of 713.6. Calcium content (128.7) was similar to the value of 137.3 from the literature. Iron content (6.62) was slightly higher than the given value of 6.05 from other studies. Zinc content (5.10) was lower than the value of 7.11 found in the literature. Manganese content (9.60) was higher than the 7.34 value reported in the literature. Copper content (1.20) was slightly lower than the reported value of 1.63 [46]. For Santhica 23 variety, the concentration of magnesium (470.11) in this experiment was higher than the report in the literature of 329.62. Potassium content (652.69) was lower than the reported value of 997.12. Calcium content (128.7) was similar to the value of 126.18 from the literature. Iron content (6.62) was lower than the given value of 9.66 from other studies. Zinc content (5.10) was close to the value of 5.15 found in the literature. Manganese content (9.60) was higher than the 7.79 value from the literature. Copper content (1.20) was close to the reported value of 1.05 [46]. Regarding the Tiborszalassi variety, the level of magnesium (322.80) in this experiment was lower than that of 410.6 reported by the literature. Potassium content (993.3) was much higher than the value of 415.1 reported. Calcium content (107.83) was lower than the value of 172.0 from the literature. Iron content (11.63) was higher than the given value of 9.70 from other studies. Zinc content (5.38) was lower than the value of 8.46 found in the literature. Manganese content (12.48) was higher than the 8.46 level from the literature. Copper content (1.08) was slightly lower than the reported value of 1.76 [46]. All these variations may be due to factors like soil composition and fertilization [20,31,38].

When compared to the nutrient reference values (NRVs) of minerals established by the European Union [47] while considering the average of the mineral content of the 29 whole hempseed varieties and a serving size of 30 g, hempseeds are nutritionally interesting for human consumption (Figure 9). Manganese was the only mineral that could completely fulfill and exceed the NRV, and magnesium, copper and iron were able to cover more than 25% of the NRV. In addition, whole hempseeds could be considered a sodium-free food because this mineral was less than 5 mg/100 g [48].

### 4.5. Cannabinoid Content

Cannabinoids in industrial hempseeds had been reported to be very low [21,49] because their synthesis occurs in glandular trichomes that are not present in the seeds so their content is associated to a contamination from vegetative materials [38]. The cannabinoids present in industrial hemp are under strong environmental influences [50]. Even though cannabinoids are present at very small quantities within all the industrial hempseed varieties, as reported in the literature [33], CBD concentrations were the highest of all the cannabinoids obtained in the 29 varieties, with values that ranged from 8 to 92 µg/g. These results were higher than a previous study that reported CBD concentrations measured using the GC/MS technique with 77 commercial hempseeds for human consumption and observed a range from 0.32 to 25.55 µg/g (mean: 7.190 µg/g) [51]. Three other studies, which used the same technique and some of the hemp varieties herein analyzed, reported CBD values ranging from 1.125 to 2.039% [50], 1.0 to 1.27% [33], and 1.03 to 1.87% [52]. We can consider that these values were higher than ours because of the nature of the samples; the first study used the third upper part of the plant (including flowers, seeds, and leaves) and the second and third study used the flowering tops. In addition, the Helena variety had a higher CBD content than the Marina and Fedora 17 varieties as reported in the literature [52].

THC content leads the legislation of industrial hempseeds [37]. In accordance with the limit of 0.2%, this study found that 24 varieties did not contain any THC. From the five varieties that showed a small amount of THC, four of them did not have EU Registration. In fact, in the same studies discussed above, THC levels were also under the legal limit. The 77 commercial hemp seeds showed lower THC concentrations that ranged from 0.06 to 5.91 µg/g (mean: 0.89 µg/g). The other studies, which used the top of the plant and the flowers, reported ranges of 0.137 to 0.581% [50], 0.05 to 0.07% [33], and 0.08 to 0.10% [52], respectively. A comparison of Futura 75 variety seeds could be made with a study on this variety that reported a CBD content of 561 µg/g and a THC level of 212 µg/g, but the technique used by that study was HPLC-MS [35]. These values were higher than the ones we found in the same variety, which were 9 and 0 µg/g for CBD and THC, respectively.

## 5. Conclusions

A complete and comprehensive composition study of 29 varieties of whole industrial hempseeds is presented. They contain interesting amounts of fat and protein. Hempseed varieties analyzed in this work contain high amounts of PUFA and the primary fatty acids detected were linoleic acid and α-linolenic acid. Moreover, the amino acid profile of hempseeds constitutes a good source of essential amino acids, and it was found that arginine and glutamic acid were the most abundant amino acids. Regarding the cannabinoid content, CBD and THC were present at very low values. Interestingly, the Tisza variety was balanced for crude protein, fat, and also amino acid content. However, more studies are needed to quantify the presence of antinutritional compounds, phenolic compounds, and bioactive peptides. To conclude, the nutritional composition of hempseeds with hull makes them suitable to be added into the diets of humans or animals as a highly beneficial novel ingredient. Moreover, knowing that the chemical composition and the presence of bioactive compounds of cultivated plants is strongly influenced by the environment in which they grow, we recommend the performance of similar studies in other countries in order to have a wider approach regarding this topic.

## Figures and Tables

**Figure 1 foods-13-00210-f001:**
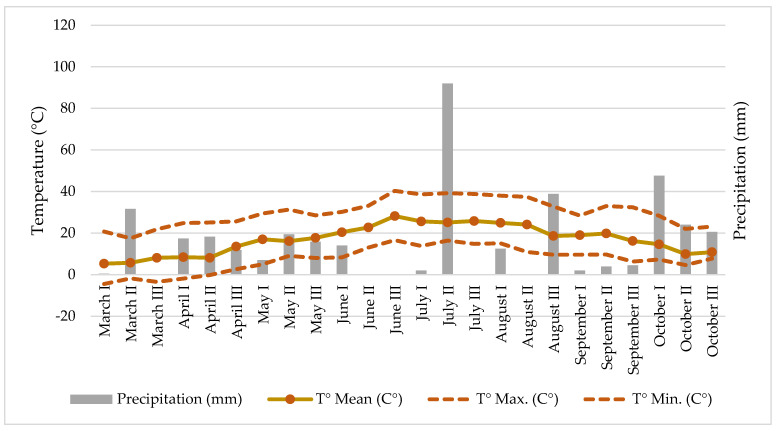
Climate conditions (precipitation and temperature) at the hemp field.

**Figure 2 foods-13-00210-f002:**
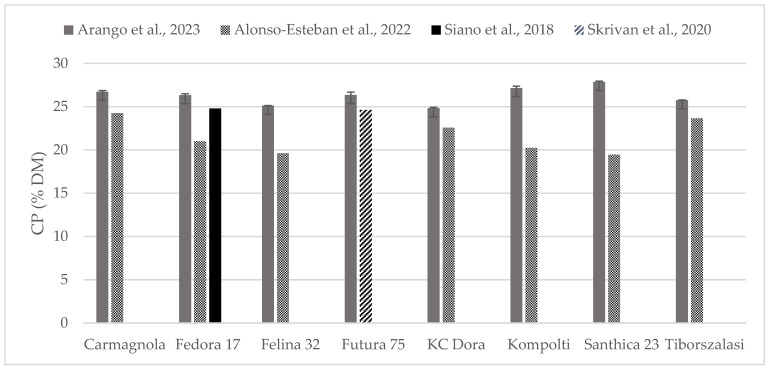
Content of crude protein (% DM) in whole hempseeds of Carmagnola, Fedora 17, Felina 32, Futura 75, KC Dora, Kompolti, Santhica 23, and Tiborszalassi varieties in comparison with the literature [27,31,32].

**Figure 3 foods-13-00210-f003:**
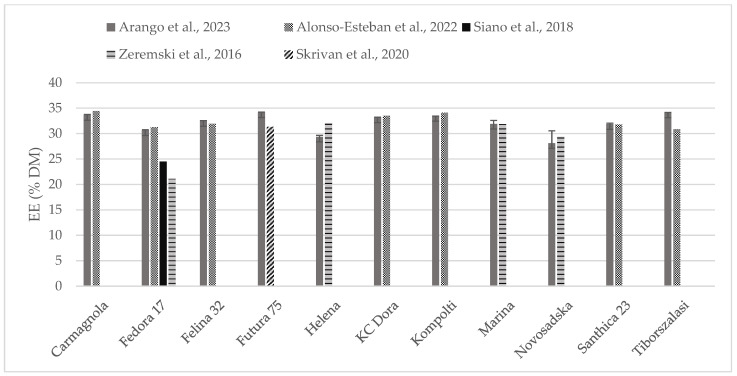
Content of fat (% DM) in whole hempseeds of Carmagnola, Fedora 17, Felina 32, Futura 75, Helena, KC Dora, Kompolti, Marina, Novosadska, Santhica 23, and Tiborszalassi varieties in comparison with the literature [27,31,32,33].

**Figure 4 foods-13-00210-f004:**
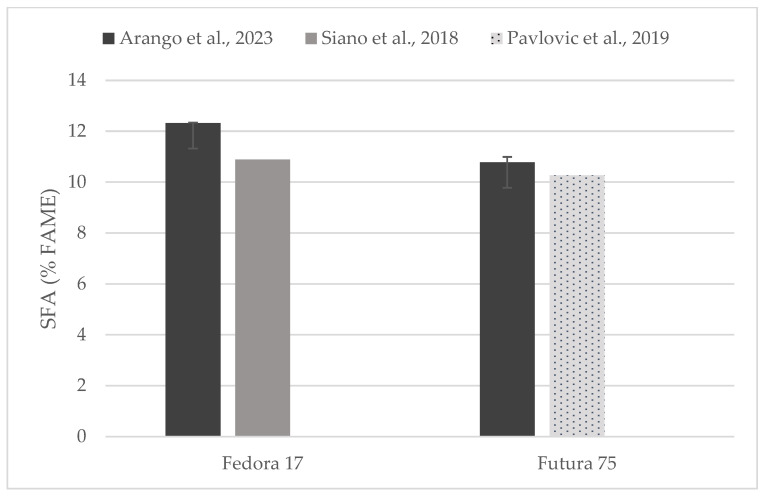
Content of saturated fatty acids (% of FAME) in seeds of Fedora 17 and Futura 75 varieties in comparison with the literature [31,35].

**Figure 5 foods-13-00210-f005:**
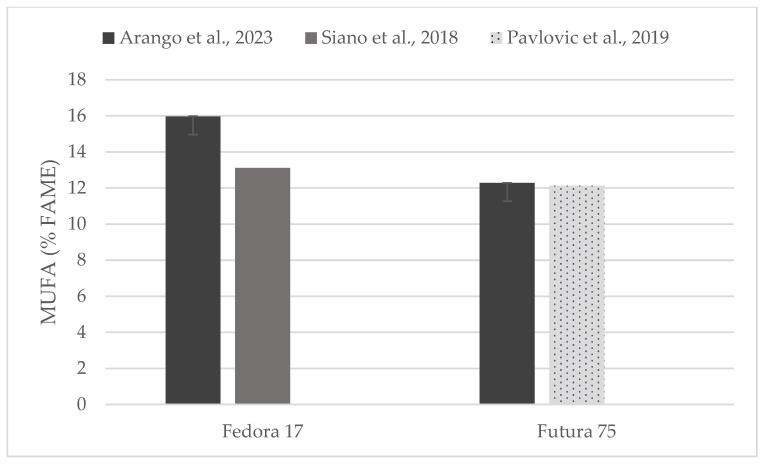
Content of mono-unsaturated fatty acids (% of FAME) in seeds of Fedora 17 and Futura 75 varieties in comparison with the literature [31,35].

**Figure 6 foods-13-00210-f006:**
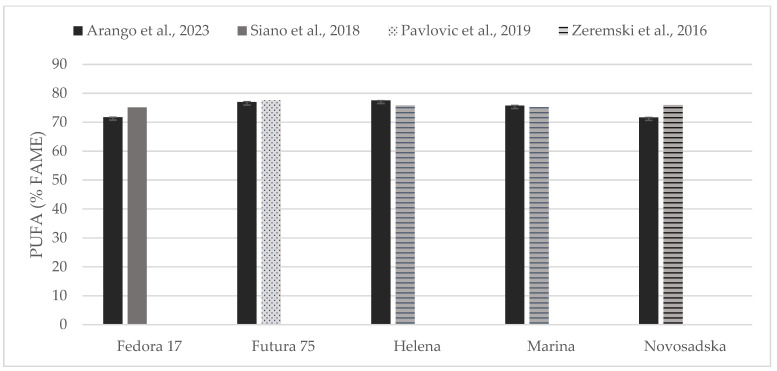
Content of poly unsaturated fatty acids (% of FAME) in seeds of Fedora 17, Futura 75, Helena, Marina, and Novosadska varieties in comparison with the literature [31,33,35].

**Figure 7 foods-13-00210-f007:**
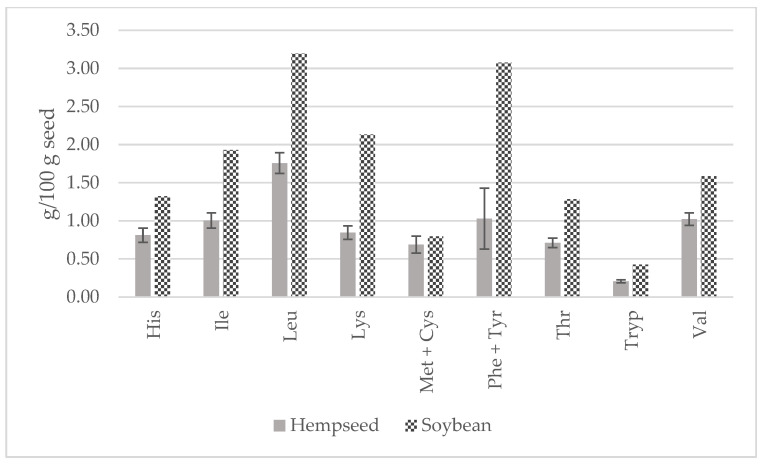
Contents of essential amino acids (g/100 g of seed) in whole hempseeds and soybean.

**Figure 8 foods-13-00210-f008:**
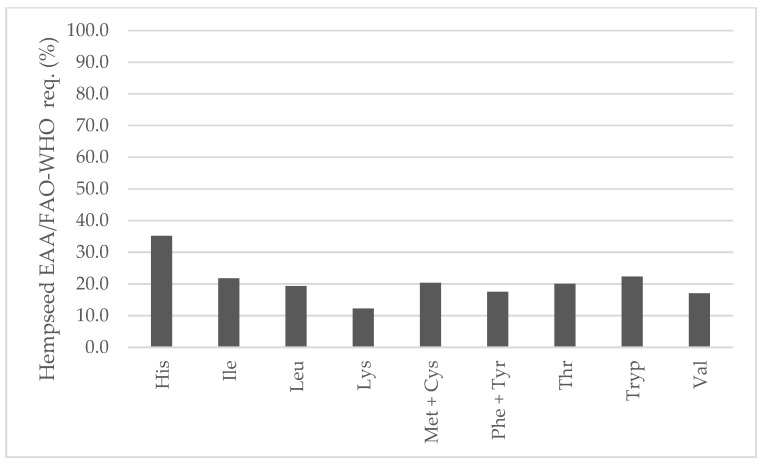
Contribution (%) to the FAO/WHO suggested essential amino acids requirements for a 70 kg adult from the habitual serving size (30 g) of whole hempseeds.

**Figure 9 foods-13-00210-f009:**
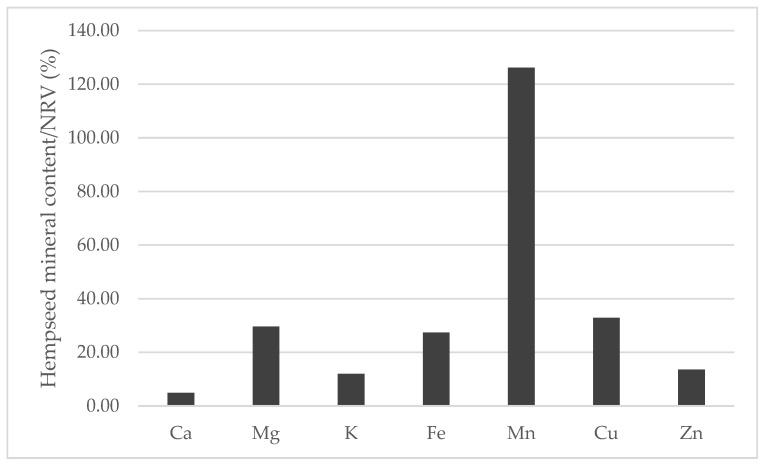
Contribution (%) to the nutrient reference values (NRV) of minerals in the habitual serving size (30 g) of whole hempseeds.

**Table 1 foods-13-00210-t001:** Industrial hempseed varieties analyzed and their main characteristics.

Variety	Origin	Sex	EU Registration
Antal	Hungary	Dioecious	No
Bacalmas	Hungary	Dioecious	No
Carmagnola	Italy	Dioecious	Yes
Chameleon	Holland	Monoecious	Yes
Dioica 88	France	Dioecious	Yes
Epsilon 88	France	Monoecious	Yes
Fedora 17	France	Monoecious	Yes
Felina 32	France	Monoecious	Yes
Ferimon FR 8194	France	Monoecious	Yes
Fibrol	Hungary	Monoecious	Yes
Futura 75	France	Monoecious	Yes
Helena	Serbia	Monoecious	Yes
KC Dora	Hungary	Monoecious	Yes
KC Virtus	Hungary	Dioecious	Yes
KC Zuzana	Hungary	Monoecious	Yes
Kina	China	Dioecious	No
Kompolti	Hungary	Dioecious	Yes
Lovrin110	Romania	Dioecious	Yes
Marina	Serbia	Dioecious	Yes
Monoica	Hungary	Monoecious	Yes
Novosadska	Serbia	Dioecious	No
Novosadska+	Serbia	Dioecious	No
Santhica 23	France	Monoecious	Yes
Secuieni jubileu	Romania	Monoecious	Yes
Silesia	Poland	Monoecious	No
Simba	Serbia	Dioecious	No
Tiborszallasi	Hungary	Dioecious	Yes
Tisza	Hungary	Dioecious	Yes
Wojko	Poland	Monoecious	Yes

**Table 2 foods-13-00210-t002:** Nutritional composition (%DM) of 29 hempseed varieties.

Variety	DM	CP	EE	CHO	Ash
Antal	92.87 ± 0.05 ^kl^	23.04 ± 0.01 ^no^	32.09 ± 0.05 ^g–i^	31.26 ± 0.05 ^e–g^	6.47 ± 0.07 ^b^
Bacalmas	92.63 ± 0.16 ^mn^	25.38 ± 0.27 ^e–h^	28.91 ± 0.13 ^o–q^	32.25 ± 0.56 ^de^	6.08 ± 0.01 ^cd^
Carmagnola	93.36 ± 0.06 ^bc^	26.73 ± 0.13 ^cd^	33.61 ± 0.15 ^c–e^	27.84 ± 0.28 ^j–l^	5.18 ± 0.06 ^g–i^
Chameleon	93.10 ± 0.02 ^f–h^	25.67 ± 0.02 ^ef^	31.83 ± 0.01 ^h–j^	30.71 ± 0.05 ^gh^	4.88 ± 0.04 ^jk^
Dioica 88	93.29 ± 0.05 ^b–d^	25.07 ± 0.17 ^gh^	29.85 ± 0.11 ^m–o^	33.38 ± 0.27 ^cd^	4.97 ± 0.04 ^ij^
Epsilon 88	92.36 ± 0.07 ^o^	22.56 ± 0.03 ^o^	21.12 ± 0.14 ^r^	43.00 ± 0.04 ^a^	5.67 ± 0.07 ^e^
Fedora 17	92.54 ± 0.04 ^n^	26.34 ± 0.17 ^d^	30.63 ± 0.12 ^k–m^	30.96 ± 0.05 ^e–h^	4.60 ± 0.04 ^ml^
Felina 32	92.94 ± 0.06 ^i–k^	25.13 ± 0.02 ^f–h^	32.45 ± 0.09 ^f–h^	30.31 ± 0.01 ^gh^	5.05 ± 0.05 ^h–j^
Ferimon FR8194	93.52 ± 0.05 ^a^	26.54 ± 0.04 ^d^	34.97 ± 0.27 ^ab^	26.47 ± 0.37 ^mn^	5.53 ± 0.01 ^ef^
Fibrol	93.06 ± 0.09 ^g–i^	25.46 ± 0.09 ^e–g^	35.67 ± 0.05 ^a^	26.38 ± 0.13 ^mn^	5.54 ± 0.01 ^fe^
Futura 75	93.40 ± 0.01 ^ab^	26.36 ± 0.32 ^d^	34.16 ± 0.07 ^bc^	26.72 ± 0.01 ^l–n^	6.16 ± 0.25 ^c^
Helena	90.76 ± 0.01 ^p^	25.55 ± 1.28 ^e–g^	29.36 ± 0.29 ^n–p^	29.62 ± 1.74 ^hi^	6.24 ± 0.19 ^c^
KC Dora	93.17 ± 0.01 ^d–g^	24.84 ± 0.09 ^h–j^	33.15 ± 0.07 ^d–f^	28.94 ± 0.14 ^ij^	6.24 ± 0.02 ^c^
KC Virtus	92.76 ± 0.07 ^lm^	23.33 ± 0.18 ^l–n^	33.04 ± 0.07 ^e–g^	28.90 ± 0.10 ^ij^	7.49 ± 0.06 ^a^
KC Zuzana	93.13 ± 0.02 ^e–h^	23.23 ± 0.18 ^mn^	31.55 ± 0.13 ^h–k^	32.70 ± 0.32 ^d^	5.64 ± 0.01 ^e^
Kina	93.24 ± 0.07 ^c–e^	25.00 ± 0.25 ^g–i^	34.24 ± 0.25 ^bc^	28.60 ± 0.59 ^ij^	5.40 ± 0.03 ^fg^
Kompolti	92.91 ± 0.02 ^jk^	27.15 ± 0.23 ^c^	33.44 ± 0.02 ^c–f^	26.08 ± 0.65 ^mn^	6.23 ± 0.37 ^c^
Lovrin110	93.00 ± 0.06 ^h–k^	26.85 ± 0.01 ^cd^	31.06 ± 0.06 ^j–l^	29.84 ± 0.06 ^hi^	5.24 ± 0.06 ^gh^
Marina	90.34 ± 0.01 ^q^	23.79 ± 0.48 ^cd^	31.91 ± 0.67 ^h–j^	28.51 ± 1.31 ^i–k^	6.13 ± 0.14 ^cd^
Monoica	92.93 ± 0.06 ^i–k^	21.63 ± 0.14 ^p^	31.69 ± 0.03 ^h–j^	34.25 ± 0.09 ^c^	5.36 ± 0.03 ^fg^
Novosadska	93.00 ± 0.05 ^h–k^	23.89 ± 0.07 ^kl^	28.12 ± 2.43 ^q^	36.26 ± 2.41 ^b^	4.73 ± 0.01 ^kl^
Novosadska+	93.04 ± 0.06 ^g–j^	24.47 ± 0.13 ^i–k^	30.18 ± 0.01 ^l–n^	33.10 ± 0.01 ^cd^	5.29 ± 0.07 ^g^
Santhica 23	93.20 ± 0.14 ^e–d^	27.86 ± 0.10 ^b^	31.85 ± 0.20 ^h–j^	28.20 ± 0.33 ^jk^	5.29 ± 0.11 ^g^
Secuieni jubileu	92.93 ± 0.07 ^i–k^	25.38 ± 0.01 ^e–h^	32.03 ± 0.10 ^h–j^	30.80 ± 0.03 ^f–h^	4.72 ± 0.06 ^kl^
Silesia	92.92 ± 0.04 ^i–k^	24.38 ± 0.03 ^jk^	31.44 ± 0.23 ^i–k^	32.15 ± 0.39 ^d–f^	4.96 ± 0.09 ^ij^
Simba	92.59 ± 0.05 ^n^	25.15 ± 0.02 ^f–h^	35.01 ± 0.25 ^ab^	25.87 ± 0.28 ^mn^	6.56 ± 0.01 ^b^
Tiborszalassi	92.99 ± 0.08 ^h–k^	25.75 ± 0.03 ^e^	34.13 ± 0.04 ^b–d^	27.20 ± 0.17 ^k–m^	5.91 ± 0.19 ^d^
Tisza	92.87 ± 0.01 ^kl^	28.92 ± 0.05 ^a^	33.91 ± 0.36 ^c–e^	25.49 ± 0.27 ^n^	4.55 ± 0.03 ^lm^
Wojko	93.27 ± 0.16 ^b–e^	24.47 ± 0.18 ^i–k^	28.48 ± 0.19 ^pq^	35.92 ± 0.66 ^b^	4.40 ± 0.14 ^m^
Mean ± SD	92.83 ± 0.68	25.17 ± 1.61	31.72 ± 2.85	30.40 ± 3.82	5.54 ± 0.72
SEM	0.0699	0.2877	0.4917	0.6687	0.1124
Soybean	89.91 ± 0.01	41.79 ± 0.18	18.65 ± 0.17	34.36 ± 0.50	5.20 ± 0.16

DM: dry matter; CP: crude protein; EE: ether extract; CHO: total carbohydrates. Means with different superscripts are significantly different at *p* < 0.05.

**Table 3 foods-13-00210-t003:** Fatty acid profiles (% of FAME) of 29 hempseed varieties.

	16:0	16:1	18:0	18:1	18:2n−6	18:3n−6	18:3n−3	20:0	20:2n−6	22:0	24:0
Antal	7.33 ± 0.01 ^h^	0.17 ± 0.05 ^a–d^	3.08 ± 0.01 ^c–e^	15.55 ± 0.01 ^i^	54.77 ± 0.01 ^q^	2.09 ± 0.01 ^fg^	15.02 ± 0.01 ^n^	0.82 ± 0.01 ^c–f^	0.63 ± 0.01 ^e–j^	0.40 ± 0.01 ^c–f^	0.17 ± 0.01 ^c–g^
Bacalmas	7.71 ± 0.01 ^d^	0.11 ± 0.01 ^ef^	3.14 ± 0.01 ^bc^	16.10 ± 0.01 ^f^	54.51 ± 0.01 ^s^	2.10 ± 0.01 ^fg^	14.39 ± 0.01 ^r^	0.79 ± 0.01 ^e–h^	0.58 ± 0.01 ^i–k^	0.40 ± 0.01 ^c–f^	0.18 ± 0.01 ^c–f^
Carmagnola	7.09 ± 0.01 ^lm^	0.11 ± 0.02 ^d–f^	2.61 ± 0.01 ^n^	14.12 ± 0.01 ^m^	55.17 ± 0.01 ^n^	1.59 ± 0.01 ^m^	17.54 ± 0.01 ^g^	0.68 ± 0.20 ^m–l^	0.59 ± 0.01 ^h–k^	0.35 ± 0.01 ^e–g^	0.15 ± 0.01 ^d–i^
Chameleon	7.31 ± 0.02 ^hi^	0.12 ± 0.01 ^c–f^	3.40 ± 0.01 ^a^	14.29 ± 0.01 ^l^	56.62 ± 0.01 ^c^	2.57 ± 0.01 ^e^	13.38 ± 0.01 ^v^	0.99 ± 0.01 ^a^	0.66 ± 0.01 ^e–g^	0.47 ± 0.01 ^a–c^	0.19 ± 0.01 ^c–e^
Dioica 88	6.81 ± 0.01 ^q^	0.17 ± 0.07 ^a–c^	2.37 ± 0.01 ^p^	11.82 ± 0.01 ^v^	55.99 ± 0.01 ^g^	1.25 ± 0.01 ^p^	20.19 ± 0.01 ^a^	0.61 ± 0.01 ^o^	0.51 ± 0.01 ^l^	0.17 ± 0.22 ^h^	0.11 ± 0.15 ^hi^
Epsilon 88	7.91 ± 0.01 ^c^	0.15 ± 0.01 ^a–f^	3.35 ± 0.01 ^a^	15.64 ± 0.08 ^h^	55.12 ± 0.01 ^o^	2.13 ± 0.01 ^f^	13.33 ± 0.01 ^v^	0.89 ± 0.01 ^b^	0.69 ± 0.01 ^e^	0.55 ± 0.09 ^a^	0.26 ± 0.01 ^ab^
Fedora 17	8.05 ± 0.01 ^b^	0.16 ± 0.02 ^a–e^	2.73 ± 0.03 ^jk^	15.80 ± 0.01 ^g^	54.63 ± 0.01 ^r^	3.56 ± 0.01 ^c^	12.62 ± 0.01 ^x^	0.87 ± 0.01 ^bc^	0.91 ± 0.01 ^c^	0.45 ± 0.01 ^b–d^	0.22 ± 0.01 ^bc^
Felina 32	7.18 ± 0.01 ^k^	0.15 ± 0.01 ^a–f^	2.70 ± 0.01 ^j–l^	13.91 ± 0.01 ^o^	56.00 ± 0.01 ^g^	2.63 ± 0.01 ^e^	15.35 ± 0.01 ^m^	0.76 ± 0.01 ^g–j^	0.78 ± 0.01 ^d^	0.37 ± 0.01 ^d–g^	0.17 ± 0.01 ^c–g^
Ferimon FR8194	6.95 ± 0.01 ^o^	0.14 ± 0.01 ^b–f^	2.67 ± 0.01 ^lm^	11.72 ± 0.01 ^w^	56.45 ± 0.01 ^d^	3.95 ± 0.01 ^a^	15.72 ± 0.01 ^k^	0.71 ± 0.01 ^j–m^	1.18 ± 0.01 ^a^	0.37 ± 0.02 ^d–g^	0.14 ± 0.01 ^e–i^
Fibrol	7.47 ± 0.01 ^g^	0.20 ± 0.07 ^a^	3.04 ± 0.01 ^e^	16.73 ± 0.01 ^c^	55.81 ± 0.01 ^hi^	1.43 ± 0.01 ^n^	13.60 ± 0.01 ^u^	0.85 ± 0.01 ^b–d^	0.41 ± 0.01 ^m^	0.36 ± 0.01 ^d–g^	0.16 ± 0.01 ^d–h^
Futura 75	6.41 ± 0.03 ^s^	0.13 ± 0.01 ^b–f^	3.11 ± 0.11 ^b–d^	12.15 ± 0.01 ^t^	56.25 ± 0.01 ^e^	2.05 ^g^ ± 0.01 ^h^	17.73 ± 0.01 ^f^	0.73 ± 0.09 ^j–l^	0.76 ± 0.01 ^d^	0.39 ± 0.02 ^c–g^	0.14 ± 0.02 ^e–i^
Helena	7.00 ± 0.01 ^n^	0.20 ± 0.01 ^a^	2.60 ± 0.01 ^n^	11.65 ± 0.07 ^x^	55.50 ± 0.01 ^k^	2.00 ± 0.01 ^hi^	19.20 ± 0.01 ^b^	0.60 ± 0.01 ^op^	0.80 ± 0.14 ^d^	0.30 ± 0.01 ^g^	0.10 ± 0.01 ^i^
KC Dora	6.79 ± 0.01 ^q^	0.12 ± 0.01 ^c–f^	2.82 ± 0.01 ^i^	12.81 ± 0.01 ^s^	55.81 ± 0.01 ^h^	1.87 ± 0.01 ^j^	17.89 ± 0.01 ^e^	0.73 ± 0.01 ^i–l^	0.68 ± 0.01 ^ef^	0.35 ± 0.01 ^e–g^	0.14 ± 0.01 ^e–i^
KC Virtus	7.09 ± 0.01 ^lm^	0.12 ± 0.01 ^c–f^	2.91 ± 0.01 ^gh^	11.88 ± 0.07 ^u^	56.29 ± 0.01 ^e^	1.85 ± 0.01 ^jk^	17.93 ± 0.01 ^e^	0.80 ± 0.01 ^d–g^	0.65 ± 0.01 ^e–h^	0.33 ± 0.07 ^fg^	0.16 ± 0.01 ^d–h^
KC Zuzana	7.12 ± 0.01 ^l^	0.12 ± 0.01 ^c–f^	2.95 ± 0.01 ^fg^	17.81 ± 0.01 ^a^	52.79 ± 0.01 ^u^	1.80 ± 0.01 ^jk^	15.53 ± 0.01 ^l^	0.72 ± 0.01 ^j–m^	0.61 ± 0.01 ^g–k^	0.38 ± 0.01 ^c–g^	0.17 ± 0.01 ^c–g^
Kina	6.91 ± 0.01 ^p^	0.18 ± 0.07 ^ab^	2.74 ± 0.01 ^j^	11.16 ± 0.01 ^y^	57.13 ± 0.01 ^a^	0.55 ± 0.01 ^r^	20.24 ± 0.01 ^a^	0.60 ± 0.01 ^op^	0.21 ± 0.01 ^o^	0.20 ± 0.01 ^h^	0.12 ± 0.01 ^g–i^
Kompolti	6.36 ± 0.01 ^t^	0.11 ± 0.01 ^ef^	2.68 ± 0.01 ^k–m^	13.39 ± 0.01 ^r^	55.25 ± 0.01 ^m^	1.44 ± 0.01 ^n^	19.08 ± 0.01 ^c^	0.64 ± 0.01 ^no^	0.57 ± 0.01 ^j–l^	0.33 ± 0.01 ^fg^	0.14 ± 0.01 ^e–i^
Lovrin110	7.66 ± 0.01 ^e^	0.12 ± 0.01 ^c–f^	3.35 ± 0.02 ^a^	15.59 ± 0.01 ^hi^	53.59 ± 0.01 ^t^	1.79 ± 0.01 ^k^	15.54 ± 0.01 ^l^	0.99 ± 0.01 ^a^	0.62 ± 0.01 ^f–j^	0.50 ± 0.01 ^ab^	0.26 ± 0.01 ^ab^
Marina	6.30 ± 0.01 ^u^	0.10 ± 0.01 ^f^	3.05 ± 0.07 ^e^	13.50 ± 0.01 ^q^	56.05 ± 0.07 ^f^	0.75 ± 0.07 ^q^	18.70 ± 0.14 ^d^	0.75 ± 0.07 ^g–j^	0.30 ± 0.01 ^n^	0.40 ± 0.01 ^c–f^	0.20 ± 0.01 ^cd^
Monoica	7.08 ± 0.01 ^m^	0.12 ± 0.01 ^c –f^	3.13 ± 0.01 ^bc^	16.66 ± 0.01 ^d^	55.25 ± 0.01 ^m^	1.35 ± 0.07 ^o^	14.81 ± 0.06 ^p^	0.69 ± 0.01 ^k–n^	0.43 ± 0.01 ^m^	0.35 ± 0.01 ^e–g^	0.13 ± 0.01 ^f –i^
Novosadska	7.73 ± 0.07 ^d^	0.13 ± 0.01 ^b–f^	3.16 ± 0.01 ^b^	15.78 ± 0.01 ^g^	54.67 ± 0.01 ^r^	1.61 ± 0.01 ^lm^	14.93 ± 0.01 ^o^	0.84 ± 0.01 ^b–e^	0.43 ± 0.01 ^m^	0.51 ± 0.07 ^ab^	0.22 ± 0.01 ^bc^
Novosadska+	7.56 ± 0.01 ^f^	0.10 ± 0.01 ^f^	2.85 ± 0.01 ^i^	15.20 ± 0.07 ^j^	55.12 ± 0.01 ^o^	1.45 ± 0.01 ^n^	15.92 ± 0.01 ^j^	0.73 ± 0.01 ^i–l^	0.55 ± 0.07 ^kl^	0.37 ± 0.01 ^d–g^	0.16 ± 0.01 ^d–h^
Santhica 23	7.01 ± 0.01 ^n^	0.15 ± 0.02 ^a–e^	2.51 ± 0.01 ^o^	11.08 ± 0.01 ^z^	56.94 ± 0.09 ^b^	2.98 ± 0.01 ^d^	17.24 ± 0.01 ^h^	0.67 ± 0.01 ^mn^	1.02 ± 0.01 ^b^	0.32 ± 0.01 ^fg^	0.15 ± 0.01 ^d–i^
Secuieni jubileu	7.18 ± 0.01 ^k^	0.14 ± 0.01 ^b–f^	2.64 ± 0.01 ^mn^	16.94 ± 0.01 ^b^	56.00 ± 0.01 ^g^	1.45 ± 0.14 ^n^	13.87 ± 0.01 ^t^	0.86 ± 0.01 ^bc^	0.31 ± 0.01 ^n^	0.44 ± 0.01 ^b–e^	0.20 ± 0.01 ^cd^
Silesia	7.29 ± 0.01 ^ij^	0.14 ± 0.01 ^b–f^	2.97 ± 0.07 ^f^	16.53 ± 0.01 ^e^	55.77 ± 0.01 ^i^	1.79 ± 0.01 ^k^	14.04 ± 0.01 ^s^	0.56 ± 0.07 ^p^	0.44 ± 0.01 ^m^	0.33 ± 0.01 ^fg^	0.15 ± 0.01 ^d–i^
Simba	7.27 ± 0.01 ^j^	0.12 ± 0.01 ^c–f^	2.86 ± 0.01 ^hi^	14.02 ± 0.01 ^n^	55.02 ± 0.01 ^p^	1.66 ± 0.01 ^l^	17.20 ± 0.01 ^h^	0.71 ± 0.01 ^j–m^	0.64 ± 0.01 ^e–i^	0.35 ± 0.01 ^e–g^	0.15 ± 0.01 ^d–i^
Tiborszalassi	6.74 ± 0.01 ^r^	0.12 ± 0.01 ^c–f^	2.70 ± 0.01 ^j–l^	13.78 ± 0.05 ^p^	55.69 ± 0.01 ^j^	1.99 ± 0.01 ^hi^	17.10 ± 0.01 ^i^	0.74 ± 0.01 ^h–k^	0.68 ± 0.01 ^ef^	0.32 ± 0.05 ^fg^	0.14 ± 0.01 ^e–i^
Tisza	7.07 ± 0.01 ^m^	0.15 ± 0.03 ^a–f^	2.92 ± 0.01 ^fg^	15.57 ± 0.01 ^i^	55.71 ± 0.01 ^j^	1.97 ± 0.01 ^i^	14.68 ± 0.01 ^q^	0.80 ± 0.01 ^d–g^	0.59 ± 0.01 ^h–k^	0.40 ± 0.01 ^c–f^	0.13 ± 0.03 ^f–h^
Wojko	8.13 ± 0.01 ^a^	0.14 ± 0.01 ^b–f^	3.06 ± 0.01 ^de^	14.48 ± 0.01 ^k^	55.45 ± 0.01 ^l^	3.68 ± 0.02 ^b^	12.73 ± 0.01 ^w^	0.78 ± 0.01 ^f–i^	0.88 ± 0.01 ^c^	0.35 ± 0.01 ^e–g^	0.31 ± 0.01 ^a^
Mean ± SD	7.19 ± 0.46	0.14 ± 0.03	2.90 ± 0.26	14.33 ± 1.92	55.50 ± 0.92	1.98 ± 0.78	16.05 ± 2.20	0.75 ± 11	0.62 ± 0.21	0.37 ± 0.09	0.17 ± 0.05
SEM	0.0157	0.0264	0.0271	0.0286	0.0207	0.0325	0.0342	0.0247	0.0294	0.0475	0.0285
Soybean	9.90 ± 0.01	0.00	4.90 ± 0.01	26.90 ± 0.01	52.00 ± 0.01	0.00	5.35 ± 0.07	0.40 ± 0.01	0.00	0.30 ± 0.01	0.20 ± 0.01

Means with different superscripts are significantly different at *p* < 0.05.

**Table 4 foods-13-00210-t004:** Groups of fatty acids (% of FAME) of 29 hempseed varieties.

	Σ SFA	Σ MUFA	Σ PUFA	ω6/ω3
Antal	11.77 ± 0.04 ^gh^	15.71 ± 0.05 ^hi^	72.52 ± 0.01 ^p^	3.83 ± 0.01 ^l^
Bacalmas	12.22 ± 0.01 ^f^	16.21 ± 0.01 ^f^	71.57 ± 0.01 ^tu^	3.98 ± 0.01 ^h^
Carmagnola	10.89 ± 0.02 ^k^	14.23 ± 0.02 ^m^	74.88 ± 0.01 ^j^	3.27 ± 0.01 ^st^
Chameleon	12.37 ± 0.01 ^de^	14.41 ± 0.01 ^l^	73.22 ± 0.01 ^m^	4.47 ± 0.01 ^c^
Dioica 88	10.07 ± 0.07 ^o^	11.99 ± 0.07 ^u^	77.94 ± 0.01 ^b^	2.86 ± 0.01 ^x^
Epsilon 88	12.95 ± 0.08 ^a^	15.78 ± 0.08 ^h^	71.27 ± 0.01 ^v^	4.35 ± 0.01 ^d^
Fedora 17	12.32 ± 0.02 ^ef^	15.96 ± 0.02 ^g^	71.72 ± 0.01 ^s^	4.68 ± 0.01 ^b^
Felina 32	11.17 ± 0.02 ^j^	14.06 ± 0.01 ^o^	74.77 ± 0.02 ^k^	3.87 ± 0.01 ^j^
Ferimon FR 8194	10.84 ± 0.01 ^k^	11.86 ± 0.01 ^v^	77.30 ± 0.01 ^d^	3.92 ± 0.01 ^i^
Fibrol	11.82 ± 0.07 ^g^	16.93 ± 0.07 ^c^	71.25 ± 0.01 ^v^	4.24 ± 0.01 ^e^
Futura 75	10.78 ± 0.21 ^kl^	12.28 ± 0.01 ^t^	76.94 ± 0.21 ^e^	3.33 ± 0.01 ^r^
Helena	10.70 ± 0.01 ^lm^	11.80 ± 0.01 ^v^	77.50 ± 0.01 ^c^	3.04 ± 0.01 ^v^
KC Dora	10.83 ± 0.01 ^k^	12.93 ± 0.01 ^s^	76.24 ± 0.01 ^g^	3.26 ± 0.01 ^t^
KC Virtus	11.28 ± 0.07 ^ij^	12.00 ± 0.07 ^u^	76.72 ± 0.01 ^f^	3.28 ± 0.01 ^s^
KC Zuzana	11.34 ± 0.01 ^i^	17.93 ± 0.01 ^a^	70.73 ± 0.01 ^w^	3.56 ± 0.01 ^o^
Kina	10.52 ± 0.07 ^n^	11.34 ± 0.07 ^w^	78.13 ± 0.01 ^b^	2.86 ± 0.01 ^x^
Kompolti	10.16 ± 0.01 ^o^	13.50 ± 0.01 ^r^	76.34 ± 0.01 ^g^	3.00 ± 0.01 ^w^
Lovrin110	12.76 ± 0.01 ^b^	15.71 ± 0.01 ^i^	71.53 ± 0.01 ^u^	3.60 ± 0.01 ^n^
Marina	10.65 ± 0.07 ^m^	13.60 ± 0.07 ^q^	75.75 ± 0.07 ^h^	3.05 ± 0.02 ^u^
Monoica	11.38 ± 0.01 ^i^	16.78 ± 0.01 ^d^	71.84 ± 0.01 ^r^	3.85 ± 0.01 ^k^
Novosadska	12.46 ± 0.01 ^d^	15.90 ± 0.01 ^g^	71.64 ± 0.01 ^st^	3.80 ± 0.01 ^m^
Novosadska+	11.66 ± 0.01 ^h^	15.30 ± 0.07 ^j^	73.04 ± 0.07 ^n^	3.59 ± 0.01 ^n^
Santhica 23	10.59 ± 0.11 ^mn^	11.23 ± 0.02 ^x^	78.18 ± 0.09 ^a^	3.54 ± 0.01 ^p^
Secuieni jubileu	11.30 ± 0.04 ^i^	17.08 ± 0.01 ^b^	71.62 ± 0.04 ^tu^	4.16 ± 0.04 ^f^
Silesia	11.30 ± 0.01 ^i^	16.67 ± 0.01 ^e^	72.03 ± 0.01 ^q^	4.13 ± 0.01 ^g^
Simba	11.33 ± 0.01 ^i^	14.14 ± 0.01 ^n^	74.53 ± 0.01 ^l^	3.33 ± 0.01 ^r^
Tiborszalassi	10.63 ± 0.05 ^mn^	13.91 ± 0.05 ^p^	75.46 ± 0.01 ^i^	3.41 ± 0.01 ^q^
Tisza	11.32 ± 0.03 ^i^	15.71 ± 0.03 ^hi^	72.97 ± 0.01 ^n^	3.97 ± 0.01 ^h^
Wojko	12.63 ± 0.02 ^c^	14.62 ± 0.01 ^k^	72.75 ± 0.02 ^o^	4.71 ± 0.01 ^a^
Mean ± SD	11.38 ± 0.78	14.47 ± 1.92	74.15 ± 2.48	3.69 ± 0.52
SEM	0.0586	0.0367	0.0480	0.0085
Soybean	15.75 ± 0.07	26.90 ± 0.01	57.35 ± 0.07	9.72 ± 0.13

Σ SFA: Total saturated fatty acids; Σ MUFA: total mono-unsaturated fatty acids; Σ PUFA: total polyunsaturated fatty acids. Means with different superscripts are significantly different at *p* < 0.05.

**Table 5 foods-13-00210-t005:** Amino acid composition (g/100 g of protein) of 29 hempseed varieties.

	Ala	Arg	Asp	Cys	Glu	Gly	His *	Ile *	Leu *	Lys *	Met *	Phe *	Pro	Ser	Thr *	Trp *	Tyr	Val *
Antal	3.92 ^a–d^	15.79 ^fg^	9.58 ^a–g^	1.58 ^d–j^	13.91 ^c–f^	3.77 ^a–h^	3.41 ^e–g^	4.10 ^i–k^	7.62 ^c–g^	3.75 ^b–d^	1.21 ^h–m^	3.48 ^d^	6.53 ^d–f^	4.49 ^f–j^	3.12 ^c–e^	0.91 ^c–f^	2.64 ^ij^	4.46 ^c–g^
Bacalmas	3.52 ^hi^	14.69 ^kl^	9.40 ^d–j^	1.60 ^c–i^	14.21 ^b–d^	3.84 ^a–f^	3.25 ^g–j^	3.91 ^kl^	7.10 ^i–k^	3.34 ^g–j^	1.76 ^bc^	0.03 ^h^	6.36 ^e–h^	4.34 ^h–j^	2.93 ^e–h^	0.84 ^e–g^	2.45 ^jk^	4.27 ^g–k^
Carmagnola	3.78 ^c–g^	15.28 ^hi^	9.03 ^ml^	1.39 ^j–m^	13.08 ^hi^	3.62 ^g–j^	3.46 ^d–g^	4.23 ^e–j^	7.50 ^e–h^	3.58 ^c–g^	1.05 ^mn^	2.46 ^g^	6.65 ^cd^	4.43 ^h–j^	2.98 ^d–h^	0.86 ^c–g^	2.82 ^f–i^	4.25 ^g–k^
Chameleon	3.61 ^f–i^	14.43 ^lm^	9.64 ^a–e^	1.71 ^a–e^	14.23 ^b–d^	3.67 ^c–j^	3.08 ^jk^	4.15 ^g–k^	7.15 ^i–k^	3.34 ^g–j^	1.89 ^b^	4.20 ^a^	6.64 ^c–e^	4.63 ^c–h^	2.86 ^f–h^	0.87 ^c–g^	2.74 ^g–j^	4.21 ^h–k^
Dioica 88	3.68 ^d–i^	16.02 ^ef^	9.37 ^e–k^	1.61 ^c–h^	14.27 ^b–d^	3.96 ^a^	3.68 ^a–d^	4.57 ^bc^	7.88 ^bc^	3.76 ^b–d^	1.65 ^cd^	0.01 ^h^	6.48 ^d–f^	4.72 ^b–g^	3.16 ^b–e^	0.86 ^d–g^	3.26 ^a–c^	4.38 ^d–j^
Epsilon 88	3.72 ^c–g^	12.66 ^p^	9.76 ^ab^	1.84 ^ab^	13.49 ^e–i^	3.73 ^c–i^	3.36 ^f–i^	4.40 ^b–g^	7.85 ^bc^	3.44 ^f–j^	0.90 ^n^	0.00 ^h^	8.76 ^a^	4.43 ^h–j^	2.85 ^gh^	1.14 ^ab^	3.06 ^c–f^	4.60 ^b–d^
Fedora 17	3.87 ^a–e^	17.56 ^a^	9.74 ^a–c^	1.35 ^k–m^	14.47 ^bc^	3.82 ^a–g^	3.85 ^ab^	4.65 ^ab^	7.84 ^bc^	3.89 ^ab^	1.66 ^cd^	0.02 ^h^	6.88 ^bc^	4.88 ^a–d^	3.37 ^ab^	0.81 ^e–g^	3.01 ^c–g^	4.73 ^ab^
Felina 32	3.58 ^g–i^	15.39 ^hi^	9.15 ^i–m^	1.27 ^m^	13.78 ^d–g^	3.54 ^ij^	3.38 ^f–h^	4.14 ^g–k^	7.39 ^f–i^	3.53 ^c–h^	1.03 ^mn^	0.03 ^h^	5.73 ^m–o^	4.30 ^ij^	3.06 ^c–h^	0.84 ^e–g^	2.47 ^jk^	4.39 ^d–j^
Ferimon FR 8194	3.51 ^hi^	16.41 ^d^	9.26 ^h–l^	1.47 ^f–l^	14.24 ^b–d^	3.67 ^c–j^	3.76 ^a–c^	4.57 ^bc^	7.67 ^b–f^	3.52 ^d–i^	2.27 ^a^	3.12 ^f^	5.39 ^pq^	4.61 ^d–h^	3.14 ^b–e^	0.83 ^e–g^	3.42 ^ab^	4.42 ^c–i^
Fibrol	3.79 ^c–g^	17.50 ^a^	9.82 ^a^	1.55 ^e–k^	14.84 ^ab^	3.87 ^a–d^	3.87 ^a^	4.50 ^b–e^	7.66 ^b–f^	3.96 ^ab^	1.42 ^e–h^	0.00 ^h^	6.45 ^d–g^	5.18 ^a^	3.41 ^a^	0.83 ^e–g^	3.18 ^b–d^	4.36 ^e–k^
Futura 75	3.87 ^a–e^	15.53 ^gh^	9.21 ^i–l^	1.40 ^i–m^	13.33 ^f–i^	3.65 ^e–j^	3.66 ^a–d^	4.47 ^b–e^	7.61 ^c–g^	3.59 ^c–f^	1.30 ^f–l^	3.06 ^f^	6.26 ^f–i^	4.56 ^e–i^	3.09 ^c–g^	0.84 ^e–g^	3.21 ^b–d^	4.58 ^b–e^
Helena	3.59 ^g–i^	14.15 ^mn^	9.34 ^e–l^	1.65 ^b–g^	13.72 ^d–i^	3.68 ^c–j^	3.07 ^jk^	3.72 ^l^	6.98 ^j–l^	3.74 ^b–d^	1.35 ^f–k^	4.26 ^a^	6.57 ^de^	4.43 ^g–j^	2.94 ^e–h^	1.00 ^b–d^	2.49 ^jk^	3.96 ^l^
KC Dora	3.51 ^hi^	16.36 ^d^	9.23 ^i–l^	1.64 ^b–g^	14.22 ^b–d^	3.87 ^a–e^	3.63 ^b–e^	4.25 ^e–j^	7.21 ^h–j^	3.78 ^bc^	1.23 ^h–m^	3.66 ^c^	5.64 ^m–p^	4.47 ^f–j^	3.07 ^c–h^	0.83 ^e–g^	2.48 ^jk^	4.14 ^kl^
KC Virtus	3.46 ^i^	15.18 ^ij^	9.24 ^i–l^	1.89 ^a^	13.87 ^c–f^	3.68 ^c–j^	3.14 ^i–k^	4.13 ^h–k^	7.58 ^c–g^	3.48 ^e–j^	1.58 ^c–e^	0.00 ^h^	5.66 ^m–p^	4.23 ^j^	2.84 ^h^	0.87 ^c–g^	3.25 ^a–d^	4.19 ^i–l^
KC Zuzana	4.04 ^ab^	15.76 ^fg^	9.23 ^i–l^	1.42 ^h–m^	13.67 ^d–i^	3.74 ^b–i^	3.47 ^d–g^	4.35 ^c–i^	7.97 ^b^	3.71 ^b–e^	1.14 ^k–m^	0.00 ^h^	5.82 ^l–n^	4.62 ^d–h^	3.22 ^a–c^	0.79 ^fg^	3.16 ^b–e^	4.61 ^b–d^
Kina	3.94 ^a–c^	16.48 ^cd^	9.56 ^a–h^	1.35 ^k–m^	14.52 ^bc^	3.89 ^a–c^	3.82 ^ab^	4.53 ^b–d^	7.57 ^c–g^	4.11 ^a^	1.19 ^j–m^	0.03 ^h^	7.14 ^b^	4.72 ^b–g^	3.15 ^b–e^	0.85 ^d–g^	2.80 ^f–i^	4.31 ^f–k^
Kompolti	3.80 ^b–g^	16.74 ^bc^	9.26 ^g–l^	1.25 ^m^	13.72 ^d–i^	3.82 ^a–g^	3.85 ^ab^	4.56 ^b–d^	7.51 ^d–h^	3.90 ^ab^	1.28 ^g–l^	0.00 ^h^	5.11 ^q^	4.88 ^a–d^	3.12 ^c–e^	0.82 ^e–g^	2.79 ^f–i^	4.25 ^g–k^
Lovrin110	3.70 ^c–i^	15.80 ^fg^	9.31 ^f–l^	1.63 ^c–g^	13.76 ^d–h^	3.69 ^c–j^	3.45 ^d–g^	4.15 ^g–k^	7.31 ^g–i^	3.74 ^b–d^	1.20 ^i–m^	0.00 ^h^	5.46 ^op^	4.41 ^h–j^	2.98 ^d–h^	0.81 ^fg^	2.47 ^jk^	4.27 ^g–k^
Marina	3.56 ^g–i^	14.08 ^n^	9.31 ^f–l^	1.79 ^a–c^	13.96 ^c–f^	3.66 ^d–j^	3.00 ^k^	3.68 ^l^	6.86 ^kl^	3.24 ^j^	1.11 ^l–n^	3.89 ^b^	5.91 ^j–m^	4.43 ^h–j^	2.85 ^gh^	1.18 ^a^	2.48 ^jk^	3.98 ^l^
Monoica	3.84 ^b–f^	14.74 ^k^	9.82 ^a^	1.55 ^e–k^	14.28 ^b–d^	3.87 ^a–e^	3.42 ^e–g^	4.18 ^f–j^	7.87 ^bc^	3.32 ^h–j^	1.16 ^k–m^	4.18 ^a^	6.73 ^cd^	4.81 ^b–e^	2.98 ^d–h^	1.23 ^a^	3.04 ^c–f^	4.46 ^c–g^
Novosadska	3.63 ^e–i^	13.59 ^o^	9.06 ^k–m^	1.78 ^a–d^	13.04 ^i^	3.59 ^h–j^	3.03 ^jk^	4.07 ^jk^	6.79 ^l^	3.42 ^f–j^	1.49 ^d–f^	3.31 ^e^	6.04 ^i–l^	4.25 ^j^	2.86 ^f–h^	0.97 ^c–e^	2.26 ^k^	4.18 ^j–l^
Novosadska+	3.77 ^c–g^	14.89 ^jh^	9.14 ^j–m^	1.67 ^b–f^	13.30 ^f–i^	3.64 ^f–j^	3.36 ^f–i^	4.33 ^c–j^	7.53 ^d–g^	3.58 ^c–g^	1.51 ^d–f^	0.00 ^h^	6.17 ^g–j^	4.37 ^h–j^	3.02 ^c–h^	0.90 ^c–f^	3.14 ^b–e^	4.51 ^b–f^
Santhica 23	3.48 ^hi^	14.96 ^jk^	9.15 ^i–m^	1.51 ^e–k^	13.16 ^g–i^	3.48 ^j^	3.18 ^h–k^	4.14 ^h–k^	7.61 ^c–g^	3.42 ^f–j^	1.16 ^k–m^	0.00 ^h^	6.28 ^f–i^	4.35 ^h–j^	2.84 ^h^	0.82 ^e–g^	3.06 ^c–f^	4.42 ^c–h^
Secuieni jubileu	3.85 ^b–f^	16.30 ^de^	9.60 ^a–f^	1.35 ^k–m^	14.11 ^c–e^	3.80 ^a–h^	3.59 ^c–f^	4.38 ^c–h^	7.74 ^b–e^	3.61 ^c–f^	1.40 ^e–j^	2.98 ^f^	6.49 ^d–f^	4.61 ^d–h^	3.09 ^c–f^	0.84 ^e–g^	2.96 ^d–h^	4.46 ^c–g^
Silesia	3.48 ^hi^	14.15 ^mn^	9.42 ^d–j^	1.65 ^b–g^	13.77 ^d–g^	3.85 ^a–f^	3.26 ^g–j^	4.17 ^f–j^	7.15 ^i–k^	3.28 ^ij^	1.14 ^k–m^	0.05 ^h^	5.56 ^n–p^	4.40 ^h–j^	2.83 ^h^	1.02 ^bc^	2.23 ^k^	4.29 ^f–k^
Simba	3.79 ^b–g^	17.04 ^b^	9.43 ^c–j^	1.28 ^lm^	15.49 ^a^	3.85 ^a–f^	3.78 ^a–c^	4.49 ^b–e^	7.62 ^c–g^	3.73 ^b–e^	1.41 ^e–i^	3.09 ^f^	6.11 ^h–k^	4.75 ^b–f^	3.10 ^c–f^	0.89 ^c–f^	2.87 ^e–i^	4.45 ^c–g^
Tiborszalassi	3.78 ^c–g^	16.78 ^bc^	9.46 ^b–i^	1.28 ^lm^	13.93 ^c–f^	3.95 ^ab^	3.74 ^a–c^	4.43 ^b–f^	7.82 ^c–d^	4.12 ^a^	1.47 ^d–g^	0.00 ^h^	5.84 ^k–n^	4.94 ^ab^	3.20 ^a–d^	0.72 ^g^	2.69 ^h–j^	4.62 ^bc^
Tisza	3.72 ^c–h^	15.81 ^fg^	8.89 ^m^	1.45 ^g–l^	13.29 ^f–i^	3.64 ^f–j^	3.51 ^d–f^	4.30 ^d–j^	7.37 ^f–i^	3.60 ^c–f^	2.14 ^a^	2.97 ^f^	6.17 ^g–j^	4.49 ^f–j^	2.96 ^d–h^	0.77 ^fg^	3.06 ^c–f^	4.31 ^f–k^
Wojko	4.11 ^a^	16.86 ^b^	9.71 ^a–d^	1.40 ^i–m^	14.21 ^b–d^	3.78 ^a–h^	3.51 ^d–f^	4.87 ^a^	8.34 ^a^	3.37 ^f–j^	1.50 ^d–f^	0.01 ^h^	6.91 ^bc^	4.93 ^a–c^	3.22 ^a–c^	0.82 ^e–g^	3.50 ^a^	4.87 ^a^
Mean ± SD	3.72 ± 0.19	15.55 ± 1.19	9.38 ± 0.27	1.53 ± 0.19	13.93 ± 0.59	3.74 ± 0.14	3.47 ± 0.28	4.29 ± 0.28	7.52 ± 0.36	3.62 ± 0.25	1.40 ± 0.33	1.55 ± 1.76	6.37 ± 0.69	4.57 ± 0.25	3.04 ± 0.18	0.90 ± 0.13	2.86 ± 0.36	4.38 ± 0.22
SEM	0.12	0.15	0.15	0.10	0.34	0.11	0.11	0.13	0.15	0.12	0.11	0.08	0.14	0.14	0.12	0.08	0.14	0.11
Soybean	3.71	10.98	8.95	0.74	12.43	3.44	3.52	4.22	8.49	5.67	1.37	4.72	8.70	4.30	3.42	1.13	3.46	5.13

* Essential amino acids. Ala: alanine; Arg: arginine; Asp: asparagine; Cys: cysteine; Glu: glutamine; Gly: glycine; His: histidine; Ile: isoleucine; Leu: leucine; Lys: lysine; Met: methionine; Phe: phenylalanine; Pro: proline; Ser: serine; Thr: threonine; Trp: tryptophan; Tyr: tyrosine; Val: valine. Means with different superscripts are significantly different at *p* < 0.05.

**Table 6 foods-13-00210-t006:** Amino acid composition (g/100 g of seed) of 29 hempseed varieties.

	Ala	Asp	Arg	Cys	Glu	Gly	His *	Ile *	Leu *	Lys *	Met *	Phe *	Pro	Ser	Thr *	Trp *	Tyr	Val *
Antal	0.84 ^f–i^	2.05 ^mn^	3.38 ^m^	0.34 ^d–h^	2.98 ^ml^	0.81 ^ij^	0.73 ^k–n^	0.88 ^k–m^	1.63 ^jk^	0.80 ^h–k^	0.26 ^hi^	0.75 ^e–g^	1.40 ^hi^	0.96 ^i–k^	0.67 ^i–l^	0.20 ^d–g^	0.56 ^m–o^	0.95 ^i–l^
Bacalmas	0.83 ^g–j^	2.22 ^g–i^	3.46 ^k^	0.38 ^a–e^	3.35 ^d–h^	0.91 ^c–g^	0.77 ^i–k^	0.92 ^i–l^	1.67 ^h–j^	0.79 ^i–l^	0.41 ^bc^	0.01 ^j^	1.50 ^g^	1.02 ^g–i^	0.69 ^f–j^	0.20 ^d–g^	0.58 ^l–n^	1.01 ^f–i^
Carmagnola	0.94 ^a–c^	2.25 ^e–h^	3.81 ^hi^	0.35 ^c–g^	3.27 ^f–j^	0.90 ^c–g^	0.86 ^d–g^	1.06 ^c–e^	1.87 ^b–d^	0.89 ^b–e^	0.26 ^hi^	0.61 ^i^	1.66 ^b^	1.11 ^ef^	0.74 ^b–g^	0.21	0.70 ^e–h^	1.06 ^d–f^
Chameleon	0.86 ^d–h^	2.31 ^b–e^	3.45 ^kl^	0.41 ^a^	3.40 ^c–f^	0.88 ^d–g^	0.74 ^j–m^	0.99 ^e–h^	1.71 ^g–i^	0.80 ^h–l^	0.45 ^b^	1.00 ^a^	1.59 ^c–e^	1.11 ^de^	0.68 ^g–k^	0.21 ^c–e^	0.65 ^g–j^	1.01 ^f–i^
Dioica 88	0.86 ^e–h^	2.20 ^h–k^	3.76 ^i^	0.38 ^a–d^	3.35 ^d–h^	0.93 ^a–d^	0.86 ^d–g^	1.07 ^b–d^	1.85 ^b–e^	0.88 ^d–f^	0.39 ^cd^	0.00 ^j^	1.52 ^fg^	1.11 ^ef^	0.74 ^b–g^	0.20 ^d–g^	0.76 ^b–e^	1.03 ^e–g^
Epsilon 88	0.78 ^jk^	2.03 ^mn^	2.64 ^q^	0.38 ^a–d^	2.81 ^n^	0.78 ^j^	0.70 ^l–o^	0.92 ^j–l^	1.63 ^jk^	0.72 ^m–o^	0.19 ^j^	0.00 ^j^	1.82 ^a^	0.92 ^k^	0.59 ^m^	0.24 ^a–c^	0.64 ^i–l^	0.96 ^i–k^
Fedora 17	0.94 ^a–c^	2.37 ^ab^	4.27 ^a^	0.33 ^e–h^	3.52 ^a–c^	0.93 ^a–d^	0.94 ^a–c^	1.13 ^ab^	1.91 ^ab^	0.95 ^ab^	0.40 ^bc^	0.00 ^j^	1.67 ^b^	1.19 ^a–c^	0.82 ^a^	0.20 ^e–g^	0.73 ^d–f^	1.15 ^ab^
Felina 32	0.84 ^f–i^	2.14 ^j–l^	3.60 ^j^	0.30 ^h^	3.22 ^h–j^	0.83 ^h–j^	0.79 ^h–j^	0.97 ^g–j^	1.73 ^gh^	0.83 ^f–i^	0.24 ^ij^	0.01 ^j^	1.34 ^i–k^	1.01 ^h–j^	0.72 ^e–i^	0.21 ^d–f^	0.58 ^k–n^	1.02 ^e–h^
Ferimon FR 8194	0.87 ^d–h^	2.30 ^b–e^	4.07 ^c^	0.37 ^a–f^	3.53 ^a–c^	0.91 ^c–f^	0.93 ^a–c^	1.13 ^ab^	1.90 ^bc^	0.87 ^ef^	0.56 ^a^	0.77 ^de^	1.34 ^i–l^	1.14 ^b–e^	0.78 ^a–d^	0.21 ^d–f^	0.85 ^a^	1.10 ^b–d^
Fibrol	0.90 ^b–f^	2.33 ^a–d^	4.15 ^b^	0.37 ^a–f^	3.52 ^a–c^	0.92 ^b–e^	0.92 ^a–d^	1.07 ^cd^	1.82 ^d–f^	0.94 ^a–c^	0.34 ^d–f^	0.00 ^j^	1.53 ^e–g^	1.23 ^a^	0.81 ^a^	0.20 ^e–g^	0.75 ^c–e^	1.03 ^e–g^
Futura 75	0.96 ^ab^	2.28 ^c–g^	3.84 ^gh^	0.35 ^b–g^	3.30 ^e–i^	0.90 ^c–g^	0.91 ^b–d^	1.11 ^a–c^	1.88 ^b–d^	0.89 ^c–e^	0.32 ^e–g^	0.76 ^ef^	1.55 ^e–g^	1.13 ^c–e^	0.76 ^a–e^	0.21 ^d–f^	0.79 ^a–d^	1.13 ^a–c^
Helena	0.83 ^g–j^	2.17 ^i–k^	3.28 ^n^	0.38 ^a–d^	3.18 ^i–k^	0.85 ^g–i^	0.71 ^k–n^	0.86 ^lm^	1.62 ^jk^	0.87 ^e–g^	0.31 ^e–g^	0.99 ^a^	1.52 ^e–g^	1.03 ^g–i^	0.68 ^h–k^	0.23 ^a–d^	0.58 ^l–n^	0.92 ^j–l^
KC Dora	0.81 ^h–k^	2.14 ^j–l^	3.79 ^hi^	0.38 ^a–d^	3.30 ^e–i^	0.90 ^c–g^	0.84 ^f–h^	0.98 ^f–i^	1.67 ^h–j^	0.88 ^ef^	0.28 ^f–i^	0.85 ^b^	1.31 ^j–m^	1.03 ^f–h^	0.71 ^e–i^	0.20 ^e–g^	0.57 ^l–n^	0.96 ^i–k^
KC Virtus	0.75 ^k^	2.00 ^mn^	3.30 ^n^	0.41 ^a^	3.01 ^lm^	0.80 ^j^	0.68 ^m–o^	0.90 ^k–m^	1.65 ^i–k^	0.75 ^k–m^	0.34 ^de^	0.00 ^j^	1.23 ^n^	0.92 ^k^	0.62 ^lm^	0.19 ^e–g^	0.71 ^e–h^	0.91 ^k–m^
KC Zuzana	0.88 ^d–g^	2.00 ^n^	3.42 ^k–m^	0.31 ^gh^	2.97 ^l–n^	0.81 ^ij^	0.75 ^i–l^	0.94 ^h–k^	1.73 ^gh^	0.81 ^h–k^	0.25 ^i^	0.00 ^j^	1.26 ^mn^	1.00 ^h–j^	0.70 ^f–j^	0.17 ^g^	0.68 ^f–i^	1.00 ^g–i^
Kina	0.92 ^b–e^	2.23 ^f–i^	3.85 ^gh^	0.32 ^gh^	3.39 ^c–g^	0.91 ^c–f^	0.89 ^b–f^	1.06 ^cd^	1.77 ^fg^	0.96 ^a^	0.28 ^g–i^	0.01 ^j^	1.67 ^b^	1.10 ^ef^	0.74 ^c–h^	0.20 ^d–g^	0.66 ^g–j^	1.01 ^f–i^
Kompolti	0.96 ^ab^	2.34 ^a–c^	4.24 ^a^	0.32 ^gh^	3.47 ^a–d^	0.97 ^ab^	0.97 ^a^	1.15 ^a^	1.90 ^bc^	0.99 ^a^	0.32 ^e–g^	0.00 ^j^	1.29 ^k–n^	1.24 ^a^	0.79 ^a–c^	0.21 ^d–f^	0.71 ^e–h^	1.08 ^c–e^
Lovrin110	0.92 ^b–d^	2.32 ^a–e^	3.94 ^ef^	0.41 ^a^	3.44 ^b–e^	0.92 ^b–e^	0.86 ^d–g^	1.04 ^d–g^	1.82 ^d–f^	0.93 ^a–d^	0.30 ^e–h^	0.00 ^j^	1.36 ^h–j^	1.10 ^ef^	0.74 ^b–f^	0.21 ^d–f^	0.62 ^j–m^	1.07 ^de^
Marina	0.76 ^k^	2.00 ^n^	3.03 ^p^	0.38 ^a–d^	3.00 ^lm^	0.79 ^j^	0.64 ^o^	0.79 ^n^	1.48 ^l^	0.70 ^no^	0.24 ^ij^	0.84 ^bc^	1.27 ^l–n^	0.95 ^jk^	0.61 ^m^	0.26 ^a^	0.53 ^no^	0.85 ^m^
Monoica	0.77 ^jk^	1.98 ^n^	2.97 ^p^	0.31 ^gh^	2.88 ^mn^	0.78 ^j^	0.69 ^m–o^	0.84 ^mn^	1.59 ^k^	0.67 ^o^	0.23 ^ij^	0.85 ^b^	1.36 ^h–j^	0.97 ^h–k^	0.60 ^m^	0.25 ^ab^	0.61 ^j–m^	0.90 ^lm^
Novosadska	0.81 ^h–k^	2.01 ^mn^	3.02 ^p^	0.40 ^ab^	2.90 ^mn^	0.80 ^j^	0.67 ^no^	0.91 ^j–m^	1.51 ^l^	0.76 ^j–m^	0.33 ^e–g^	0.74 ^f–h^	1.34 ^i–k^	0.94 ^jk^	0.63 ^k–m^	0.22 ^b–e^	0.51 ^o^	0.93 ^j–l^
Novosadska+	0.86 ^e–h^	2.08 ^lm^	3.38 ^lm^	0.38 ^a–e^	3.02 ^k–m^	0.83 ^h–j^	0.76 ^i–k^	0.98 ^f–i^	1.71 ^g–i^	0.81 ^g–j^	0.34 ^de^	0.00 ^j^	1.40 ^hi^	0.99 ^h–k^	0.69 ^g–k^	0.21 ^d–f^	0.71 ^e–g^	1.03 ^e–g^
Santhica 23	0.90 ^b–e^	2.38 ^a^	3.89 ^fg^	0.39 ^ab^	3.42 ^b–f^	0.90 ^c–g^	0.83 ^gh^	1.07 ^b–d^	1.98 ^a^	0.89 ^c–e^	0.30 ^e–h^	0.00 ^j^	1.63 ^b–d^	1.13 ^c–e^	0.74 ^c–h^	0.22 ^b–e^	0.80 ^a–d^	1.15 ^ab^
Secuieni jubileu	0.91 ^b–e^	2.26 ^d–h^	3.84 ^gh^	0.32 ^f–h^	3.33 ^d–i^	0.90 ^c–g^	0.85 ^e–h^	1.03 ^d–g^	1.83 ^c–f^	0.85 ^e–h^	0.33 ^e–g^	0.70 ^h^	1.53 ^e–g^	1.09 ^e–g^	0.73 ^d–h^	0.20 ^e–g^	0.70 ^e–i^	1.05 ^d–g^
Silesia	0.79 ^i–k^	2.13 ^kl^	3.21 ^o^	0.37 ^a–e^	3.12 ^j–l^	0.87 ^e–h^	0.74 ^j–m^	0.95 ^h–k^	1.62 ^jk^	0.74 ^l–n^	0.26 ^hi^	0.01 ^j^	1.26 ^mn^	1.00 ^h–j^	0.64 ^j–m^	0.23 ^a–d^	0.51 ^o^	0.97 ^h–j^
Simba	0.88 ^c–g^	2.20 ^h–k^	3.97 ^de^	0.30 ^h^	3.61 ^a^	0.90 ^c–g^	0.88 ^c–f^	1.05 ^c–f^	1.77 ^e–g^	0.87 ^e–g^	0.33 ^e–g^	0.72 ^gh^	1.42 ^h^	1.11 ^de^	0.72 ^e–i^	0.21 ^c–e^	0.67 ^f–j^	1.04 ^e–g^
Tiborszalassi	0.90 ^b–e^	2.26 ^d–h^	4.02 ^cd^	0.31 ^gh^	3.33 ^d–i^	0.95 ^a–c^	0.90 ^b–e^	1.06 ^cd^	1.87 ^b–d^	0.99 ^a^	0.35 ^de^	0.00 ^j^	1.40 ^hi^	1.18 ^a–d^	0.76 ^a–e^	0.18 ^fg^	0.64 ^h–k^	1.11 ^a–d^
Tisza	1.00 ^a^	2.39 ^a^	4.25 ^a^	0.39 ^a–c^	3.57 ^ab^	0.98 ^a^	0.94 ^ab^	1.16 ^a^	1.98 ^a^	0.97 ^a^	0.57 ^a^	0.80 ^cd^	1.66 ^bc^	1.21 ^ab^	0.80 ^ab^	0.21 ^d–f^	0.82 ^ab^	1.16 ^a^
Wojko	0.94 ^a–c^	2.21 ^g–j^	3.84 ^gh^	0.32 ^f–h^	3.24 ^g–j^	0.86 ^f–i^	0.80 ^hi^	1.11 ^a–c^	1.90 ^bc^	0.77 ^i–m^	0.34 ^de^	0.00 ^j^	1.57 ^d–f^	1.12 ^c–e^	0.73 ^c–h^	0.19 ^e–g^	0.80 ^a–c^	1.11 ^a–d^
Mean ± SD	0.87 ± 0.07	2.19 ± 0.13	3.64 ± 0.42	0.36 ± 0.04	3.26 ± 0.23	0.88 ± 0.06	0.81 ± 0.10	1.00 ± 0.10	1.76 ± 0.14	0.85 ± 0.09	0.33 ± 0.09	0.36 ± 0.41	1.46 ± 0.16	1.07 ± 0.10	0.71 ± 0.07	0.21 ± 0.02	0.67 ± 0.10	1.02 ± 0.08
SEM	0.03	0.04	0.04	0.02	0.08	0.02	0.03	0.03	0.04	0.03	0.03	0.02	0.03	0.03	0.03	0.02	0.03	0.03
Soybean	1.39	3.37	4.11	0.28	4.68	1.29	1.33	1.93	3.19	2.13	0.52	1.78	3.27	1.62	1.29	0.43	1.30	1.59

* Essential amino acids. Ala: alanine; Arg: arginine; Asp: asparagine; Cys: cysteine; Glu: glutamine; Gly: glycine; His: histidine; Ile: isoleucine; Leu: leucine; Lys: lysine; Met: methionine; Phe: phenylalanine; Pro: proline; Ser: serine; Thr: threonine; Trp: tryptophan; Tyr: tyrosine; Val: valine. Means with different superscripts are significantly different at *p* < 0.05.

**Table 7 foods-13-00210-t007:** Mineral composition (mg/100 g) of 29 hempseed varieties.

	Ca	Mg	K	Fe	Mn	Cu	Zn
Antal	153.76 ± 8.01 ^a–c^	376.19 ± 6.29 ^d–h^	976.28 ± 29.61 ^a–e^	32.37 ± 7.21 ^a^	10.51 ± 0.23 ^bc^	1.13 ± 0.01 ^c–f^	4.56 ± 0.01 ^f–j^
Bacalmas	149.72 ± 1.68 ^b–e^	347.20 ± 7.87 ^f–j^	1084.62 ± 36.27 ^ab^	20.49 ± 1.55 ^bc^	8.85 ± 0.24 ^c–g^	0.92 ± 0.01 ^fg^	3.09 ± 0.28 ^l–n^
Carmagnola	113.07 ± 1.90 ^i–l^	321.14 ± 9.43 ^h–k^	1041.25 ± 13.69 ^a–d^	16.58 ± 0.05 ^b–d^	8.01 ± 0.24 ^d–h^	1.16 ± 0.01 ^c–e^	3.05 ± 0.31 ^l–n^
Chameleon	137.15 ± 13.32 ^c–i^	288.98 ± 19.32 ^jk^	917.01 ± 37.36 ^a–f^	14.76 ± 1.11 ^c–f^	9.56 ± 0.01 ^b–f^	1.02 ± 0.05 ^e–g^	4.11 ± 1.18 ^g–l^
Dioica 88	157.87 ± 17.95 ^a–c^	421.53 ± 16.35 ^b–e^	590.25 ± 31.76 ^g–j^	8.11 ± 1.01 ^d–g^	9.78 ± 0.06 ^b–d^	1.27 ± 0.16 ^a–d^	7.02 ± 1.21 ^ab^
Epsilon 88	171.04 ± 3.26 ^ab^	413.43 ± 6.07 ^b–f^	538.96 ± 34.95 ^j^	9.37 ± 0.33 ^d–g^	8.63 ± 0.05 ^d–h^	0.90 ± 0.04 ^g^	4.90 ± 0.06 ^e–h^
Fedora 17	125.47 ± 3.07 ^e–k^	298.57 ± 41.80 ^i–k^	854.86 ± 21.53 ^b–i^	6.98 ± 0.88 ^e–g^	8.02 ± 0.16 ^d–h^	1.21 ± 0.06 ^b–e^	4.47 ± 0.02 ^f–k^
Felina 32	133.77 ± 8.06 ^c–j^	297.43 ± 13.26 ^i–k^	934.01 ± 51.52 ^a–f^	14.31 ± 2.96 ^c–g^	6.12 ± 0.54 ^i–k^	1.00 ± 0.03 ^e–g^	3.08 ± 0.24 ^l–n^
Ferimon FR8194	111.53 ± 2.42 ^j–l^	360.03 ± 4.18 ^e–i^	872.52 ± 105.82 ^b–h^	16.09 ± 8.65 ^c–e^	4.79 ± 0.08 ^k^	1.01 ± 0.18 ^e–g^	3.10 ± 0.27 ^l–n^
Fibrol	113.56 ± 16.68 ^i–l^	375.36 ± 4.26 ^d–h^	1182.65 ± 557.73 ^a^	25.68 ± 6.50 ^ab^	10.66 ± 0.76 ^a–c^	1.05 ± 0.07 ^d–g^	4.70 ± 1.36 ^f–i^
Futura 75	99.61 ± 14.03 ^l^	305.79 ± 68.42 ^i–k^	780.63 ± 250.73 ^c–j^	14.61 ± 3.37 ^c–f^	2.84 ± 0.02 ^l^	0.99 ± 0.24 ^e–g^	3.06 ± 0.0.30 ^l–n^
Helena	177.21 ± 7.55 ^a^	426.46 ± 2.84 ^b–e^	740.08 ± 38.75 ^e–j^	8.46 ± 2.68 ^d–g^	9.55 ± 0.31 ^b–g^	1.47 ± 0.06 ^a^	6.33 ± 0.89 ^b–d^
KC Dora	119.01 ± 9.51 ^g–l^	499.90 ± 5.72 ^a^	744.45 ± 23.62 ^d–j^	5.85 ± 0.02 ^fg^	9.41 ± 0.40 ^c–g^	1.10 ± 0.40 ^c–g^	5.58 ± 0.34 ^c–f^
KC Virtus	153.04 ± 15.05 ^a–d^	463.03 ± 54.20 ^ab^	578.33 ± 13.27 ^h–j^	6.46 ± 0.42 ^fg^	10.54 ± 1.25 ^bc^	1.12 ± 0.18 ^c–g^	5.93 ± 0.28 ^b–e^
KC Zuzana	151.47 ± 0.11 ^b–d^	328.92 ± 0.89 ^g–k^	1050.04 ± 21.33 ^a–c^	14.39 ± 0.02 ^c–g^	8.19 ± 0.09 ^d–h^	1.30 ± 0.01 ^a–c^	3.29 ± 0.01 ^k–n^
Kina	70.99 ± 22.02 ^m^	301.31 ± 24.04 ^i–k^	665.84 ± 232.85 ^f–j^	16.27 ± 12.93 ^b–e^	5.34 ± 1.22 ^jk^	0.57 ± 0.01 ^h^	4.96 ± 0.01 ^e–g^
Kompolti	128.73 ± 4.71 ^d–k^	470.11 ± 11.25 ^ab^	652.69 ± 102.91 ^f–j^	6.62 ± 1.13 ^fg^	9.60 ± 0.19 ^b–e^	1.20 ± 0.16 ^b–e^	5.10 ± 1.49 ^e–g^
Lovrin110	119.23 ± 5.06 ^g–l^	446.57 ± 5.04 ^a–c^	649.96 ± 31.97 ^f–j^	5.06 ± 0.61 ^g^	9.62 ± 0.67 ^b–e^	1.17 ± 0.10 ^b–e^	3.49 ± 0.29 ^j–m^
Marina	139.80 ± 22.54 ^c–g^	384.06 ± 39.80 ^c–h^	602.14 ± 47.21 ^g–j^	8.75 ± 0.70 ^d–g^	8.83 ± 0.87 ^c–g^	1.00 ± 0.18 ^e–g^	2.21 ± 0.62 ^n^
Monoica	133.60 ± 3.00 ^c–j^	362.31 ± 21.99 ^e–i^	878.12 ± 17.02 ^b–g^	19.29 ± 2.55 ^bc^	8.30 ± 0.04 ^d–h^	1.18 ± 0.02 ^b–e^	4.56 ± 0.03 ^f–j^
Novosadska	122.97 ± 24.62 ^g–l^	391.63 ± 14.55 ^c–g^	593.55 ± 149.77 ^g–j^	11.96 ± 5.04 ^c–g^	7.75 ± 2.89 ^f–i^	1.15 ± 0.16 ^c–e^	7.93 ± 0.79 ^a^
Novosadska+	138.98 ± 27.86 ^c–h^	274.66 ± 84.25 ^k^	793.87 ± 105.81 ^b–j^	14.31 ± 13.57 ^c–g^	5.81 ± 1.96 ^jk^	0.91 ± 0.01 ^fg^	2.88 ± 0.02 ^mn^
Santhica 23	126.18 ± 1.63 ^e–k^	329.62 ± 1.61 ^g–k^	997.12 ± 16.51 ^a–e^	9.66 ± 0.68 ^d–g^	7.79 ± 0.04 ^e–i^	1.05 ± 0.01 ^d–g^	5.15 ± 0.32 ^d–g^
Secuieni jubileu	110.12 ± 0.89 ^j–l^	441.95 ± 25.79 ^a–d^	601.72 ± 13.63 ^g–j^	9.34 ± 2.56 ^d–g^	11.35 ± 0.29 ^ab^	1.12 ± 0.21 ^c–g^	3.72 ± 0.03 ^h–m^
Silesia	148.25 ± 6.43 ^b–f^	328.13 ± 7.41 ^g–k^	997.58 ± 28.66 ^a–e^	14.26 ± 0.75 ^c–g^	7.71 ± 0.45 ^g–i^	1.08 ± 0.01 ^c–g^	3.53 ± 0.30 ^i–m^
Simba	123.47 ± 10.56 ^f–l^	347.45 ± 9.44 ^f–j^	805.21 ± 329.11 ^b–j^	13.21 ± 4.22 ^c–g^	6.93 ± 1.41 ^h–j^	1.04 ± 0.17 ^e–g^	3.09 ± 0.32 ^l–n^
Tiborszalassi	107.83 ± 2.37 ^kl^	322.80 ± 80.12 ^g–k^	993.30 ± 29.36 ^a–e^	11.63 ± 0.89 ^c–g^	12.48 ± 0.34 ^a^	1.08 ± 0.10 ^c–g^	5.38 ± 0.03 ^d–f^
Tisza	115.00 ± 10.58 ^h–l^	436.52 ± 70.13 ^a–d^	568.26 ± 47.18 ^ij^	7.53 ± 2.32 ^d–g^	9.20 ± 1.84 ^c–g^	1.15 ± 0.10 ^c–e^	5.99 ± 0.26 ^b–e^
Wojko	134.06 ± 6.56 ^c–j^	376.48 ± 21.44 ^d–h^	509.96 ± 2.33 ^j^	8.10 ± 0.35 ^d–g^	7.78 ± 0.41 ^e–i^	1.40 ± 0.11 ^ab^	6.70 ± 0.22 ^bc^
Mean ± SD	130.57 ± 23.80	370.26 ± 65.39	799.84 ± 216.50	12.78 ± 6.95	8.41 ± 2.14	1.09 ± 0.18	4.52 ± 1.48
SEM	12.12	33.84	145.23	4.62	0.91	0.11	0.58
Soybean	294.17 ± 51.81	347.02 ± 1.32	739.19 ± 266.04	12.15 ± 5.06	n.d	0.67 ± 0.02	2.67 ± 0.32

Ca: calcium; Mg: magnesium; K: potassium, Fe: iron; Mn: manganese; Cu: copper; Zn: zinc. Means with different superscripts are significantly different at *p* < 0.05.

**Table 8 foods-13-00210-t008:** Cannabinoid content (µg/g) in 29 hempseed varieties.

	CBD	THC	CBN
Antal	73.24 ± 5.81 ^bc^	15.21 ± 5.46 ^b^	16.40 ± 4.56 ^bc^
Bacalmas	94.15 ± 7.77 ^a^	n.d	11.56 ± 1.08 ^cd^
Carmagnola	29.42 ± 3.57 ^j–m^	n.d	18.98 ± 10.15 ^bc^
Chameleon	32.45 ± 6.41 ^i–l^	n.d	15.23 ± 4.99 ^b–d^
Dioica 88	76.72 ± 4.10 ^b^	n.d	16.23 ± 6.57 ^bc^
Epsilon 88	64.66 ± 4.62 ^cd^	n.d	15.71 ± 5.54 ^bc^
Fedora 17	18.39 ± 3.62 ^no^	n.d	n.d
Felina 32	59.58 ± 1.49 ^d–f^	16.71 ± 5.52 ^b^	12.68 ± 3.54 ^cd^
Ferimon FR8194	7.86 ± 0.60 ^pq^	n.d	16.12 ± 6.75 ^bc^
Fibrol	22.11 ± 4.10 ^mn^	n.d	n.d
Futura 75	8.90 ± 0.67 ^pq^	n.d	n.d
Helena	50.88 ± 3.18 ^fg^	n.d	17.57 ± 4.97 ^bc^
KC Dora	9.78 ± 0.26 ^o–q^	n.d	n.d
KC Virtus	52.93 ± 7.36 ^ef^	n.d	15.86 ± 5.35 ^bc^
KC Zuzana	23.51 ± 6.32 ^l–n^	n.d	10.63 ± 0.91 ^cd^
Kina	41.46 ± 3.98 ^h^	163.33 ± 45.60 ^a^	221.89 ± 33.00 ^a^
Kompolti	23.25 ± 3.55 ^mn^	n.d	n.d
Lovrin110	43.19 ± 6.32 ^gh^	n.d	17.06 ± 5.31 ^bc^
Marina	28.14 ± 3.66 ^k–m^	n.d	n.d
Monoica	35.10 ± 6.20 ^h–k^	n.d	15.50 ± 5.83 ^bc^
Novosadska	91.88 ± 4.60 ^a^	n.d	16.52 ± 6.30 ^bc^
Novosadska+	37.99 ± 5.33 ^h–j^	15.83 ± 5.78 ^b^	29.19 ± 3.89 ^b^
Santhica 23	5.66 ± 0.90 ^q^	n.d	n.d
Secuieni jubileu	61.08 ± 3.45 ^de^	n.d	17.71 ± 4.16 ^bc^
Silesia	20.50 ± 1.04 ^mn^	n.d	n.d
Simba	32.41 ± 3.83 ^j–l^	15.97 ± 6.88 ^b^	11.64 ± 1.16 ^cd^
Tiborszalassi	41.72 ± 3.59 ^h^	n.d	17.63 ± 5.99 ^bc^
Tisza	43.51 ± 2.24 ^gh^	n.d	17.00 ± 6.90 ^bc^
Wojko	16.23 ± 4.94 ^no^	n.d	n.d
Mean ± SD	39.54 ± 24.33	7.83 ± 30.79	18.31 ± 40.01
SEM	4.41	8.75	7.53

CBD: cannabidiol; THC: tetrahydrocannabidiol; CBN: cannabinol. n.d: not detected, <5.17 µg/g. Means with different superscripts are significantly different at *p* < 0.05.

## Data Availability

Data is contained within the article or Appendix A.

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
