# Peer review of "Chemical Characterization of 29 Industrial Hempseed (Cannabis sativa L.) Varieties"

_foods, 2024, doi:10.3390/foods13020210_

Round 1

Reviewer 1 Report

Comments and Suggestions for Authors

The manuscript (foods-2748685) studied the nutritional composition 29 varieties of industrial hempseeds such as proximate content, fatty acid and amino acid profile, and mineral composition, as well as the cannabinoid content. The manuscript is well written and there exists some new and useful information, however, major revisions are substantially needed before the reconsideration for publication.

1.     For Tables 2-8, please add the SD value for each table. If the authors do not perform replicated experiments, I suggest adding replicates to demonstrate verifiability.   

2.     Again, based on the data in Table 2-8, I suggest adding the PCA analysis to cluster the 29 varieties of industrial hempseeds.

3.     For Fig 2-8, please add the SD value for each figure, and if possible, the author should perform the multiple comparisons test and label the statistical significance. 

Reviewer 2 Report

Comments and Suggestions for Authors

Dear Authors, 

The paper is very interesting and delves into a very important and actual topic. The comprehensive characterization of 29 varieties of industrial hempseeds is noteworthy and adds significant value to the study.

However, I consider that some points must be addressed in order to improve the quality presentation of the paper:

1. Consider elaborating a bit on the safety profile of industrial hemp seeds, as it's a critical aspect for consumers and industries. Highlighting this aspect in the introduction or discussion section will contribute to the overall understanding of the safety assessment of hemp seed.

2. There appears to be some repetition of data between the results and discussion sections, with those presented in tables. Consider condensing repetitive information to maintain focus on unique findings and insightful interpretations. This will enhance the clarity and impact of your paper.

3. Including comparisons with other prominent industrial crops could provide a broader perspective. Discussing similarities and differences in key parameters could strengthen the paper's context and appeal to a wider audience.

Overall, your paper shows promise, and addressing these points could significantly enhance its impact and readability.

Best of luck!

Reviewer 3 Report

Comments and Suggestions for Authors

1.       The manuscript contains several grammar and style errors (e.g. line 47). It should be carefully checked before new submission. Please start with the title.

2.       Units must be separated from absolute values, please applied this style throughout the text.

3.       All abbreviations must be fully described before use (e.g. THC; line 27). please applied this style throughout the text.

4.       Please write sentences in correct time, if the investigation was already performed, past form is the correct one (e.g. 59-60).

5.       Introduction section is too short and the nutraceutical activity of hemp seeds in missing. Please address this topic properly. https://doi.org/10.1016/j.foodchem.2018.04.078

6.       A big concern is the fact that several factors may influence the nutritional composition of a crop harvested under open field conditions.

7.       In Table 1 the please clear EU registration meaning and add a code or registration voucher.

8.       Another big concern is the fact that authors just used two replicates to obtain results. How do authors justify the robustness of their results?

9.       Total carbohydrate content approach seems to be unusual. Please provide a reasonable explanation on this regard. Did authors consider ash content within claimed difference?

1.   Please add a chromatogram of fatty acid separation regarding section 2.4.

1.   Please add a chromatogram of amino acids regarding section 2.5.

1.   Please add a chromatogram of cannabinoids regarding section 2.7.

1.   Independently on these methods are widely used, please clear if each method used includes external or internal standard. Please focused on fatty acid and amino acid quantification.  

1.   It is hard to believe that two replicates were used to perform ANOVA-Tukey. If so, results should not reflect accurate significances. A reasonable explanation should be provided, otherwise the paper should be rejected.

1.   ANOVA-Tukey and statistically significant divergences should be stated by specific parameter in all cultivars studied. In present form the letters are quite confusing. The real problem is the fact that authors just performed two replicates.

1.   Authors claim that lysine is a limiting amino acid, however, their results state that some aromatic amino acids and methionine are actually found in less concentration than lysine. Please explain

1.   In my opinion the nutritional comparison made with different seeds from literature and the present work is questionable since the crops were subjected to different conditions. Then, this comparison does not prove that the cultivars reported in their investigation have better properties.

1.   The biggest weakness of their work is the number of replicates included for each cultivar. Experiments made with just two replicates do not reveal the natural variability of each cultivar studied.

Comments on the Quality of English Language

Moderate editing

Round 2

Reviewer 1 Report

Comments and Suggestions for Authors

The study is well-designed and presented. Conclusions are reasonable. 

Author Response

Dear Reviewer,

We extend our sincere gratitude for accepting this manuscript after your thoughtful review and suggestions. Your insights have significantly contributed to the enhancement of our work and we truly appreciate your time and effort.

Thank you once again for your valuable contribution.

Reviewer 3 Report

Comments and Suggestions for Authors

Dear Authors, you addressed almost all requiered issues. Please add the chromtograms included in yoir response as supllementary information. You should consider that these data are ma be useful for readers to  compare divergent chemical profiling in the same crop. I also note that yoir paper is still having finger and style errors. Please check it on more time before final acceptation. 

Comments on the Quality of English Language

Minor editing

Author Response

Dear Reviewer,

We extend our sincere gratitude for accepting this work after taking your time giving a thoughtful review and some valuable suggestions. We will be in contact with the editor in order to add all the chromatographs as supplementary material.

The authors